# Fetal loss and long-term maternal morbidity and mortality: A systematic review and meta-analysis

**Florentia Vlachou**[1⊙], **Despoina Iakovou**[1⊙], **Jahnavi Daru**[2⊙]*, **Rehan Khan**[3],
**Litha Pepas**[4], **Siobhan Quenby**[5], **Stamatina Iliodromiti**[2]

**1** Barts and the London School of Medicine and Dentistry, Queen Mary University of London, Whitechapel, London, United Kingdom, **2** Women's Health Research Unit, Institute for Population Health, Queen Mary University of London, London, United Kingdom, **3** Royal London Hospital, Department of Obstetrics & Gynaecology, Barts Health NHS Trust, London, United Kingdom, **4** Barts Centre of Reproductive Medicine, Barts NHS Trust, London, United Kingdom, **5** Division of Reproductive Health, Centre for Early Life, Warwick Medical School, University of Warwick, Coventry, United Kingdom

⊙ These authors contributed equally to this work.
* j.daru@qmul.ac.uk

**Data Availability Statement:** All relevant data are within the manuscript and its Supporting information files.

## Abstract

### Background

Evidence suggests common pathways between pregnancy losses and subsequent long-term maternal morbidity, rendering pregnancy complications an early chronic disease marker. There is a plethora of studies exploring associations between miscarriage and still-birth with long-term adverse maternal health; however, these data are inconclusive.

### Methods and findings

We systematically searched MEDLINE, EMBASE, AMED, BNI, CINAHL, and the Cochrane Library with relevant keywords and MeSH terms from inception to June 2023 (no language restrictions). We included studies exploring associations between stillbirth or miscarriage and incidence of cardiovascular, malignancy, mental health, other morbidities, and all-cause mortality in women without previous pregnancy loss. Studies reporting short-term morbidity (within a year of loss), case reports, letters, and animal studies were excluded. Study selection and data extraction were performed by 2 independent reviewers. Risk of bias was assessed using the Newcastle Ottawa Scale (NOS) and publication bias with funnel plots. Subgroup analysis explored the effect of recurrent losses on adverse outcomes. Statistical analysis was performed using an inverse variance random effects model and results are reported as risk ratios (RRs) with 95% confidence intervals (CIs) and prediction intervals (PIs) by combining the most adjusted RR, odds ratios (ORs) and hazard ratios (HRs) under the rare outcome assumption. We included 56 observational studies, including 45 in meta-analysis. There were 1,119,815 women who experienced pregnancy loss of whom 951,258 had a miscarriage and 168,557 stillbirth, compared with 11,965,574 women without previous loss. Women with a history of stillbirth had a greater risk of ischaemic heart disease (IHD) RR 1.56, 95% CI [1.30, 1.88]; $p < 0.001$, 95% PI [0.49 to 5.15]),

**Funding:** The author(s) received no specific funding for this work.

**Competing interests:** JD has been employed as a consultant to the World Health Organisation in the Department of Nutrition and Food Safety. This has had no impact on this work or the decision to submit this manuscript. The other authors have declared no competing interests.

**Abbreviations:** BAPM, British Association of Perinatal Medicine; CI, confidence interval; CKD, chronic kidney disease; ESRF, end-stage renal failure; HMIC, Healthcare Management Information Consortium; HR, hazard ratio; IHD, ischaemic heart disease; NOS, Newcastle Ottawa Scale; NTIS, National Technical Information Service; OR, odds ratio; PI, prediction interval; RR, risk ratio; T2DM, type 2 diabetes mellitus.

cerebrovascular (RR 1.71, 95% CI [1.44, 2.03], $p < 0.001$, 95% PI [1.92, 2.42]), and any circulatory/cardiovascular disease (RR 1.86, 95% CI [1.01, 3.45], $p = 0.05$, 95% PI [0.74, 4.10]) compared with women without pregnancy loss. There was no evidence of increased risk of cardiovascular disease (IHD: RR 1.11, 95% CI [0.98, 1.27], 95% PI [0.46, 2.76] or cerebrovascular: RR 1.01, 95% CI [0.85, 1.21]) in women experiencing a miscarriage. Only women with a previous stillbirth were more likely to develop type 2 diabetes mellitus (T2DM) (RR: 1.16, 95% CI [1.07 to 2.26]; $p < 0.001$, 95% PI [1.05, 1.35]). Women with a stillbirth history had an increased risk of developing renal morbidities (RR 1.97, 95% CI [1.51, 2.57], $p < 0.001$, 95% [1.06, 4.72]) compared with controls. Women with a history of stillbirth had lower risk of breast cancer (RR: 0.80, 95% CI [0.67, 0.96], p-0.02, 95% PI [0.72, 0.93]). There was no evidence of altered risk of other malignancies in women experiencing pregnancy loss compared to controls. There was no evidence of long-term mental illness risk in women with previous pregnancy losses (stillbirth: RR 1.90, 95% CI [0.93, 3.88], 95% PI [0.34, 9.51], miscarriage: RR 1.78, 95% CI [0.88, 3.63], 95% PI [1.13, 4.16]). The main limitations include the potential for confounding due to use of aggregated data with variable degrees of adjustment.

## Conclusions

Our results suggest that women with a history of stillbirth have a greater risk of future cardiovascular disease, T2DM, and renal morbidities. Women experiencing miscarriages, single or multiple, do not seem to have an altered risk.

## Author summary

### Why was this study done?

- Fetal loss is one of the most serious complications of pregnancy, with approximately 23,000,000 miscarriages and 2,000,000 stillbirths reported globally per annum.

- Emerging data suggest a relationship between pregnancy loss and subsequent development of maternal long-term illness.

- We conducted a systematic review and meta-analysis on the clinical evidence available to better understand the effect of pregnancy loss on the long-term maternal morbidity and mortality.

### What did the researchers do and find?

- Our systematic review included a total of 56 observational studies, with a total of 1,119,815 women who experienced pregnancy loss (those experiencing a miscarriage: 951,258; those experiencing a stillbirth: 168,557) and 11,965,574 controls.

- Women with a history of stillbirth were found to have an increased risk of developing ischaemic heart, cerebrovascular, circulatory, and renal disease, and type 2 diabetes mellitus (T2DM) as well as a slight decrease in the risk of breast cancer.

- There was no evidence of increased risk of developing these conditions in women with a history of miscarriage.

**What do these findings mean?**

- Our results show a possible association of stillbirth with future cardiovascular disease, T2DM, and renal disease, suggesting that pregnancy loss could be an indicator for future risk.

- Our work suggests future research focussing on long-term maternal health outcomes related to pregnancy loss is necessary.

- Our work uses aggregated data with variable degrees of adjustment. As a consequence, the potential for confounding variables with well-known risk factors for metabolic diseases such as gestational diabetes affecting the results is considerable.

## Introduction

Fetal loss is considered one of the most devastating adverse events in pregnancy and it is classified as either miscarriage or stillbirth. There is a variation in the definitions used for pregnancy loss globally. WHO define a loss occurring before 28 weeks of gestation as a miscarriage, while a stillbirth is a loss after 28 weeks in accordance with the ICD-10 (International Classification of Diseases) criteria (ICD-10) [1]. These definitions are not used universally, and are a reflection of many factors, including advances in neonatal and complex pregnancy care in addition to health system variation. For example, in the United States of America and Canada, miscarriage is defined as a pregnancy loss before 20 weeks gestation [2,3], while the United Kingdom (UK) a gestational age of 24 weeks is usually used; however, this may change in line with the recent British Association of Perinatal Medicine (BAPM) recommendations that "resuscitation can be considered between 22 to 24 weeks exceptions [4,5]." The global incidence of miscarriage is approximately 23,000,000 cases per annum, with 1 in 10 women estimated to experience fetal loss at least once in their lifetime. The risk of miscarriage per pregnancy can be as high as 15.3% [5]. Stillbirth is reported in approximately 2,000,000 women annually [1].

A large body of evidence suggests that women with a history of pregnancy losses are at risk of developing future cardiovascular, metabolic, malignant, and mental health morbidities, with an overall increase in the likelihood of mortality in the long-term [6–10]. Published reviews show that miscarriage (single or recurrent) can increase the likelihood of cardiovascular disease in the longer term [11,12].

Whether pregnancy is a "stress test" for long-term disease or common biological pathways related to circulation or inflammation underlie pregnancy losses and subsequent long-term disease is unclear. While existing data suggest an association between fetal loss and future subsequent morbidity or mortality, the literature is limited by small sample sizes and variation in study follow-up. To address this need, we conducted a systematic review and meta-analysis to investigate the risk of long-term morbidity and mortality in mothers who experienced fetal loss of any type, including recurrent losses.

## Methods

We conducted a systematic review and meta-analysis of existing literature in accordance with the recommended methods described by Cochrane and reported using the Preferred

Reporting Items for Systematic reviews and Meta-analysis (PRISMA) guidelines (see S1 PRISMA Checklist) [13]. The protocol for this review was registered prospectively (PROS-PERO ID: CRD42021225370).

## Search strategy

Electronic databases including MEDLINE, EMBASE, AMED, BNI, CINAHL, and the Cochrane Library were searched from inception to June 2023 to identify studies exploring the association of fetal loss with subsequent maternal morbidity and mortality. We used the terms "fetal loss," "miscarriage," "fetal death," "stillbirth," and "spontaneous abortion" including alternative spellings for exposure. Our outcome keywords included "cardiovascular disease," "myocardial infarction," "hypertension," "cerebrovascular disease," "stroke," "type 2 diabetes," "cancer," "malignancy," "anxiety," "depression," "mortality," and "morbidity" with relevant synonyms, alternative spellings, and MESH terms. There were no language or date restrictions applied to the searches and we only included human studies. (See Fig A in S1 Appendix for detailed strategy.) A search of the grey literature was performed by manually screening the reference list of papers included in the review and reviewing the reference list of related published systematic reviews. Additionally, we searched the System for Information on Grey Literature (OpenSIGLE), Healthcare Management Information Consortium (HMIC) database, National Technical Information Service (NTIS), and PsychEXTRA for relevant citations.

## Study selection and data extraction

Study selection by title and abstract screening was performed by 2 reviewers (FV and DI) independently, any disagreements or discrepancies in any part of the process were resolved through discussion with a third reviewer (SI or JD). Relevant citations were then selected for full-text assessment. Studies were included if they examined long-term incidence of cardiovascular, metabolic, malignant, mental health, or any other morbidity in a population of women who had experienced fetal loss. Long-term was defined as disease developing at least a year after pregnancy loss. Additionally, we included studies investigating long-term all-cause mortality after pregnancy loss. Studies were excluded if patients had a history of previous diagnosis of disease outcomes described before experiencing pregnancy loss. Studies reporting maternal morbidity following a pregnancy termination were excluded from our review. Case reports, letters, in vitro, and animal studies were also excluded, while we included cohort, case-control and cross-sectional studies were included.

Data were extracted in parallel by 2 independent reviewers (FV and DI) on study design, participant demographic characteristics, type of fetal loss (miscarriage, recurrent miscarriage, or stillbirth), the comparator population, number of women developing the disease outcomes in both the exposed and controls, association measure with point estimates and 95% confidence interval (CI) and any adjustment variables used in analysis.

## Risk of bias assessment

The Newcastle Ottawa Scale (NOS) was used to examine the methodological quality of studies included in this review in duplicate by 2 reviewers independently (FV and DI) [14]. Studies were evaluated against the NOS predefined criteria and stars were assigned for each domain. Papers with a total score of 7 to 9 were considered of good methodological quality (low risk of bias), while a total score of 4 to 6 and 0 to 3 were deemed to have a high and very high risk of bias, respectively. Publication bias was assessed using funnel plots with axis of standard error against risk ratio (RR) in outcomes with more than 10 studies.

## Statistical analysis

The exposure of interest was either miscarriage or stillbirth. Given the variability in gestational age definitions for pregnancy loss across different studies, we used the definition provided in each study by the authors. Subgroup analysis was performed for women experiencing recurrent pregnancy loss. Meta-analysis was conducted where at least 3 studies reported the same exposure-outcome combination. Data synthesis and analysis were performed using inverse variance weighted random-effects meta-analysis to combine the most adjusted reported odds ratios (ORs), RRs, and hazard ratios (HRs) to produce pooled RRs and 95% CIs under the rare outcome assumption. Additionally, we repeated the analysis using the unadjusted raw data from each study as dichotomous outcomes with Mantel–Haenszel statistical method using a random effect model.

The outcomes included ischaemic heart disease (IHD), cerebrovascular disease, overall circulatory/cardiovascular disease, type 2 diabetes mellitus (T2DM), renal disease, breast cancer, uterine cancer (endometrial and cervical), ovarian cancer, malignancy, and mental health disease. IHD included morbidity or mortality from myocardial infarction, and coronary heart disease. Cerebrovascular disease included morbidity or mortality secondary to either ischaemic or haemorrhagic stroke. Renal disease involved the diagnosis or mortality from chronic kidney disease (CKD) and end-stage renal failure (ESRF). Malignancy outcomes included diagnosis or mortality of any type of cancer. Mental illness included the diagnosis of any type of depression, anxiety, or history of substance abuse.

The pooled results were presented as RR with 95% CIs and 95% prediction intervals (PIs). Heterogeneity was assessed using $I^2$, $Tau^2$, and $Chi^2$ statistics. Analysis was performed using the Review Manager 5.4 and Python version 3.10 and the Python libraries numpy 1.23.5, statsmodel 0.14, and scipy 1.11.3.

# Results

Our search yielded a total of 11,959 citations. Following screening, 145 articles were identified for full-text assessment, after which 59 studies were included (Fig 1).

## Study characteristics

There were 59 studies included in the review, 32 of these were cohort studies of which 16 were retrospective cohorts [10,15–29] and 16 prospective cohorts [9,12,30–43]. There were 25 case-control studies [8,44–67] and 2 retrospective cross-sectional studies [68,69].

A total of 1,112,250 women had experienced at least 1 miscarriage and were compared to 6,719,014 controls. A total of 171,332 women had at least 1 stillbirth and were compared to 8,547,276 controls. Most studies were conducted in high-income regions/countries including North America (*n* = 12), Europe (*n* = 35), Australia (*n* = 2), and Israel (*n* = 6). The remaining 4 studies were conducted in Asia (Japan: *n* = 1, China: *n* = 1, India: *n* = 1, Iran: *n* = 1). The mean follow-up period for outcome assessment from index pregnancy for the miscarriage cohort was 19 while for stillbirth this was 17 years. Most of the studies did not provide a gestation cut off for the definition of stillbirth or miscarriage (37 out of 59). The remaining studies provided definitions varying between 20- and 28-weeks gestational age as the cut off in distinguishing between stillbirth and miscarriage, with only 3 specifically reporting first trimester losses. Study characteristics are presented in Table 1 for studies included in the analyses of miscarriage and Table 2 for stillbirth.

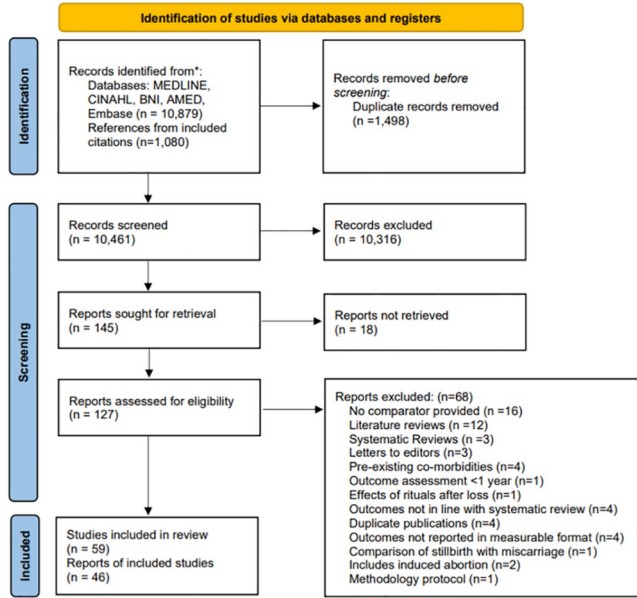

**Fig 1. PRISMA Flowchart of study inclusion and exclusion.** AMED; Allied and Complementary Medicine, BNI; British Nursing Index, CINAHL; Cumulative Index to Nursing and Allied Health Literature.

## Risk of bias assessment

In the studies of women experiencing a miscarriage ($n = 49$), the overall risk of bias was low. Specifically, among the domains of selection, comparability, and outcome/exposure, most studies received 7 out of 9 or above with more than half being awarded a score of 9. Only 1 study was identified as being at high risk of bias [37].

There were 29 studies looking at women experiencing a stillbirth with most being at low risk of bias, with only 1 study identified as being at high risk of bias [15] (Figs B–D in S1 Appendix).

Publication bias was assessed in 2 outcomes: IHD and breast cancer in the miscarriage arm. Both funnel plots appear symmetrical denoting low risk of publication bias (Fig E in S1 Appendix).

## Outcomes

**Ischaemic heart disease (IHD).** A total of 14 studies [8,12,17,18,24,26,28,41,42,44,48,54,55,63] looked at risk of IHD in women experiencing miscarriage ($n = 297,221$) compared with those with no previous history of miscarriage ($n = 1,537,851$), and 3,288 (1.12%) women with miscarriage and 18,760 (1.22%) of controls developed IHD, with a pooled RR of 1.11 (95% CI [0.98, 1.27]; $p = 0.10$, $I^2 = 89\%$, 95% PI [0.46, 2.76]) (Fig 2a). Subgroup analysis was performed to compare women who experienced recurrent miscarriage ($n = 54,489$) to those with no previous history of miscarriage ($n = 1,152,826$) and to those with a history of a single loss ($n = 200,000$). IHD was observed in 352 (0.65%), 1,081 (0.54%), and 5,222 (0.45%) women with recurrent, single, and no loss. When comparing the risk of IHD in women with recurrent losses as opposed to those without a history of fetal loss, the pooled RR was 1.21 (95% CI [0.81, 1.80]; $p = 0.35$, $I^2 = 92\%$, 95% PI [0.14, 16.33]) (Fig L in S1 Appendix). Similarly, recurrent loss compared with a single loss

**Table 1. Characteristics of all studies examining the effect of miscarriage in long-term maternal morbidity and mortality included in the systematic review.**

| Lead author and publication date (Country) | Study design | Duration of follow-up | Exposure | Gestation | Type of control | Outcome | Number of exposed with outcome | Number of controls with outcome | Result (OR, HR, RR, IRR, BDI, LCU, O/E) |
|---|---|---|---|---|---|---|---|---|---|
| Adami and colleagues (1990), Norway [65] | Case-control | - | At least 1 miscarriage | Not defined | No history of miscarriage | Breast cancer | 1 loss: 65; 2 losses: 22 | 335 | 1 loss: OR 1 95% CI (0.7, 1.5); 2 losses: OR 1.3 95% CI (0.7, 2.6) |
| Andalib and colleagues (2006), Iran [38] | Prospective cohort | - | ≥3 miscarriages | Not defined | No history of miscarriage | 1: Stress 2: Depression | - | - | 1: LCU: Exposed: 254 ± 83.6 LCU: Control: 224 ± 79.6 2: BDI: Exposed: 27.6 ± BDI: 8.8, Controls: 19.4 ± 7.1 |
| Auger and colleagues (2021), Canada [35] | Prospective cohort | 29 years | At least 1 miscarriage | <20 weeks | Pregnancy ending in live birth | Mortality | 342 | 6,965 | HR 1.48 95% CI (1.39, 1.62) |
| Bergant and colleagues (1997), Austria [37] | Prospective cohort | 2 years | ≥2 miscarriages | Not defined | No history of miscarriage | 1: Anxiety 2: Somatisation 3: Life satisfaction questionnaire 4: Depression | - | - | 4: BDI: Cases: 3.5 BDI: Controls: 3 |
| Bertuccio and colleagues (2007), Italy [54] | Case-control | - | At least 1 miscarriage | Not defined | No history of miscarriage | MI | 1 loss: 110; >1 losses: 58 | 1,277 | 1 loss: OR 1.22 95% CI (0.90, 1.66); >1 losses: OR 0.92 95% CI (0.62, 1.38) |
| Braem and colleagues (2012), the Netherlands [43] | Prospective cohort | 22 years | At least 1 miscarriage | Not defined | No history of miscarriage | Ovarian, fallopian tube and primary peritoneal cancer | 1 loss: 136; 2 losses: 22; 4 losses: 10 | 665 | 1 loss: HR 0.86 95% CI (0.71, 1.05); 2 losses: HR 1.23 95% CI (0.93, 1.61); 4 losses: HR 1.99 95% CI (1.06, 3.73) |
| Brewster and colleagues (2005), UK [52] | Case-control | 10 years | At least 1 miscarriage | <12 weeks | No history of miscarriage | Breast cancer | 2,828 | 9,781 | OR 1.02 95% CI (0.88, 1.18) |
| Brinton and colleagues (1983), USA [61] | Case-control | - | At least 1 miscarriage | >16 weeks | Singleton live birth with no history of pregnancy loss | Breast cancer | 1 loss: 226; 2 losses: 64; 3 losses: 34 | 979 | 1 loss: RR 1.1 95% CI (0.9, 1.4); 2 losses: RR 1.24 95% CI (0.8, 1.9) |
| Calle and colleagues (1995), USA [46] | Case-control | 7 years | At least 1 miscarriage | <20 weeks | No history of miscarriage | Breast cancer deaths | 1 loss: 208; 2 losses: 54; 3 losses: 34 | 951 | 1 loss: RR 0.89 95% CI (0.78, 1.02); 2 losses: RR 0.74 95% CI (0.56, 0.98); 3 losses: RR 0.85 95% CI (0.6, 1.2) |
| Charach and colleagues (2018), Israel [19] | Retrospective cohort | 26 years | At least 1 miscarriage | Not defined | No history of recurrent miscarriage | 1: Ovarian cancer 2: Uterine cancer 3: Breast cancer | 1: 5; 2: 5; 3: 56 | 1: 50; 2: 57; 3: 473 | 1: OR 1.4* p-value 0.046; 2: OR 1.2* p-value 0.651; 3: OR 1.7* p-value 0.001 |
| Coleman and colleagues (2013), USA [27] | Retrospective cohort | 25 years | At least 1 miscarriage | Not defined | Singleton live birth with no history of pregnancy loss | Mortality | 9,320 | 246,252 | OR 2.815 95% CI (2.207, 3.592) |
| Cooper and colleagues (1999), USA [48] | Case-control | - | At least 1 miscarriage | Not defined | No history of miscarriage | IHD | 1 loss: 9; ≥2 losses: 8 | - | |
| Dick and colleagues (2009), Israel [56] | Case-control | 3 years | At least 1 miscarriage | Not defined | No history of miscarriage | Ovarian cancer | 1 loss: 389; 2 losses: 96; 3 losses: 61 | 1,401 | 1 loss: OR 0.97 95% CI (0.8, 1.17); 2 losses: OR 0.93 95% CI (0.75, 1.15); 3 losses: OR 0.94 95% CI (0.61, 1.46) |
| Egerup and colleagues (2020), Denmark [60] | Case-control | 40 years | History of stillbirth | Loss >22 weeks | No history of loss | T2DM | Total: 4,314; 1 loss: 3,227; 2 losses: 729; ≥3 losses: 358 | 12613 | 1 loss: OR 1.18 95% CI (1.13, 1.23); 2 losses: OD 1.38 95% CI (1.27, 1.49); ≥3 or more: 1.71 95% CI (1.53, 1.92) |

(*Continued*)

**Table 1.** (Continued)

| Lead author and publication date (Country) | Study design | Duration of follow-up | Exposure | Gestation | Type of control | Outcome | Number of exposed with outcome | Number of controls with outcome | Result (OR, HR, RR, IRR, BDI, LCU, O/E) |
|---|---|---|---|---|---|---|---|---|---|
| *Erlandsson and colleagues (2003), Sweden* [50] | Case-control | 18 years | At least 1 miscarriage | Not defined | No history of miscarriage | Breast cancer | 1 loss: 186 <br> 2 losses: 42 | 1,531 | 1 loss: OR 0.96 95% CI (0.77, 1.19) <br> 2 losses: OR 0.66 95% CI (0.44, 0.99) |
| *Gallagher and colleagues (2011), USA* [41] | Prospective cohort | 11 years | At least 1 miscarriage | Not defined | No history of miscarriage | 1: Death from IHD <br> 2: Death from IS <br> 3: Death from HS | **1:** <br> 1 loss: 51 <br> 2 losses: 24 <br> **2:** <br> 1 loss:72 <br> 2 losses: 25 <br> **3:** <br> 1 loss: 224 <br> 2 losses: 69 | **1:** 350 <br> **2:** 482 <br> **3:** 1,303 | **1:** <br> 1 loss: HR 0.88 95% CI (0.66, 1.19) <br> 2 losses: HR 1.27 95% CI (0.84, 1.92) <br> **2:** <br> 1 loss: HR 0.87 95% CI (0.68, 1.12) <br> 2 losses: HR 0.9 95% CI (0.6, 1.35) <br> **3:** <br> 1 loss: HR 1.06 95% CI (0.92, 1.22) <br> 2 losses: HR 1.01 95% CI (0.8, 1.29) |
| *Goldacre and colleagues (2001), UK* [49] | Case-control | 30 years | At least 1 miscarriage | Not defined | No history of miscarriage | Breast cancer | 41 | - | O/E: 0.92 95% CI (0.66, 1.25) |
| *Horn and colleagues (a) (2019), Norway* [34] | Prospective cohort | 20 years | At least 1 miscarriage | <12 weeks | Singleton live birth with no history of pregnancy loss | 1: HTN <br> 2: Hypercholesterolaemia <br> 3: T2DM | **1:** <br> 1 loss: 2,614 <br> 2 losses: 537 <br> 3 losses: 202 <br> **2:** <br> 1 loss: 3,904 <br> ≥2 losses: 819 <br> 3 losses: 282 <br> **3:** <br> 1 loss: 547 <br> 2 losses: 106 <br> 3 losses: 51 | **1:** 13,902 <br> **2:** 20,253 <br> **3:** 2,672 | **1:** <br> 1 loss: HR 1.04 95% CI (1.00, 1.09) <br> 2 losses: HR 1.04 95% CI (0.95, 1.14) <br> 3 losses: HR 1.19 95% CI (1.03, 1.38) <br> **2:** <br> 1 loss: HR 1.08 95% CI (1.04, 1.12) <br> ≥2 losses: HR 1.15 95% CI (1.06, 1.24) <br> 3 losses: HR 1.15 95% CI (1.01, 1.3) <br> **3:** <br> 1 loss: HR 1.11 95% CI (1, 1.22) <br> 2 losses: HR 1.08 95% CI (0.88, 1.32) <br> 3 losses: HR 1.43 95% CI (1.06, 1.92) |
| *Horn and colleagues (b) (2019), Norway* [34] | Prospective cohort | 20 years | At least 1 miscarriage | >20 weeks | Singleton live birth with no history of pregnancy loss | 1: HTN <br> 2: HC <br> 3: T2DM | **1:** <br> 1 loss: 1,139 <br> 2 losses: 136 <br> **2:** <br> 1 loss: 1,594 <br> 2 losses: 206 <br> **3:** <br> 1 loss: 245 <br> 2 losses: 30 | **1:** 15,980 <br> **2:** 15,980 <br> **3:** 3,101 | **1:** <br> 1 loss: HR 1.11 95% CI (1.05, 1.19) <br> 2 losses: HR 1.13 95% CI (0.95, 1.35) <br> **2:** <br> 1 loss: HR 1.05 95% CI (1.04, 1.12) <br> 2 losses: HR 1.16 95% CI (1.01, 1.33) <br> **3:** <br> 1 loss: HR 1.18 95% CI (1.03, 1.35) <br> 2 losses: HR 1.04 95% CI (0.72, 1.51) |
| *Janssen and colleagues (2006), the Netherlands* [39] | Prospective cohort | 18 months | At least 1 miscarriage | <20 weeks | Pregnant women who were pregnant on recruitment and gave birth to a livebirth | Anxiety, depression, somatisation, OCD | - | 213 | - |

*(Continued)*

**Table 1.** (Continued)

| Lead author and publication date (Country) | Study design | Duration of follow-up | Exposure | Gestation | Type of control | Outcome | Number of exposed with outcome | Number of controls with outcome | Result (OR, OR, HR, RR, IRR, BDI, LCU, O/E) |
|---|---|---|---|---|---|---|---|---|---|
| Kessous and colleagues (2014), Israel [29] | Retrospective cohort | 24 years | At least 1 miscarriage | Not defined | No history of recurrent miscarriage | 1: Noninvasive CVD diagnostic procedures 2: Cardiac invasive diagnostic procedures | 1: 120 2: 54 | 1: 833 2: 370 | 1: OR 1.9 95% CI (1.6, 2.3) 2: OR 2.1 95% CI (1.6, 2.8) |
| Kharazmi and colleagues (2011), Germany [42] | Prospective cohort | 10.8 years | At least 1 miscarriage | Not defined | No history of miscarriage | 1: MI 2: Stroke | 1: 1 loss: 22 3 losses: 4 2: 1 loss: 20 3 losses: 1 | 1: 53 2: 86 | 1: 1 loss: HR 1.14 95% CI (0.65, 1.99) 3 losses: HR 5.06 95% CI (1.26, 20.29) 2: 1 loss: HR 0.72 95% CI (0.41, 1.25) 3 losses: HR 1.31 95% CI (0.161, 0.65) |
| Kharazmi and colleagues (2012), Germany [30] | Prospective cohort | 15.2 years | At least 1 miscarriage | Not defined | No history of miscarriage | T2DM | - | - | 1 loss: HR 1.34 95% CI (1.03, 1.74) 2 losses: HR 1.91 95% CI (0.94, 3.87) |
| Kharazmi and colleagues (2010), Germany [26] | Retrospective cohort | - | At least 1 miscarriage | Not defined | No history of miscarriage | 1: HTN 2: Angina pectoris | 1: 143 2: 92 | 1: 579 2: 371 | 1: OR 1.3 95% CI (0.6, 2.4) 2: OR 1.2 95% CI (0.8, 1.7) |
| Kolte and colleagues (2015), Denmark [69] | Retrospective cross-sectional | 3 years | At least 1 miscarriage | <12 weeks | Females actively trying to conceive | 1: Stress 2: Depression | 1: 124 2: 26 | 1: 420 2: 40 | 1: OR 1.59 95% CI (1.03, 2.44) 2: OR 5.53 95% CI (2.09, 14.61) |
| La Vecchia and colleagues (a) (1987), Italy [62] | Case-control | - | At least 1 miscarriage | Not defined | No history of miscarriage | Breast cancer | 1 loss: 143 2 losses: 91 | 874 | 1 loss: RR 0.9 95% CI (0.69, 1.15) 2 losses: RR 0.98 95% CI (0.71, 1.31) |
| La Vecchia and colleagues (b) (1987), Italy [63] | Case-control | - | At least 1 miscarriage | Not defined | No history of miscarriage | MI | 1 loss: 37 2 losses: 18 | 147 | 1 loss: RR 1.03 95% CI (0.69, 1.74) 2 losses: RR 1.08 95% CI (0.56, 2.06) |
| Laing and colleagues (1993), USA [67] | Case-control | 10 years | At least 1 miscarriage | Not defined | No history of miscarriage | Breast cancer | 90 | 299 | RR 1.3 95% CI (0.5, 3.3) |
| Lambalk and colleagues (2016), the Netherlands [33] | Prospective cohort | 15.2 years | At least 1 miscarriage | Not defined | No history of miscarriage | Breast cancer | 1 loss: 55 2 losses: 24 | 219 | 1 loss: HR 0.92 95% CI (0.68, 1.27) 2 losses: HR 1.03 95% CI (0.68, 0.164) |
| Lipworth and colleagues (1995), USA [47] | Case-control | - | At least 1 miscarriage | Not defined | No history of miscarriage | Breast cancer | 232 | 433 | OR 0.97 95% CI (0.79, 1.19) |
| Maino and colleagues (2016), the Netherlands [8] | Case-control | 3 years | At least 1 miscarriage | <22 weeks | No history of miscarriage | 1: MI 2: IS 3: Any arterial thrombosis | 1: 1 loss: 45 2 losses: 8 3 losses: 10 2: 1 loss: 44 2 losses: 10 3 losses: 5 3: 1 loss: 89 2 losses: 18 3 losses: 15 | 1: 155 2: 106 3: 261 | 1: 1 loss: OR 0.74 95% CI (0.45, 1.21) 2 losses: OR 0.55 95% CI (0.21, 1.46) 3 losses: OR 2.04 95% CI (0.71, 5.86) 2: 1 loss: OR 1.18 95% CI (0.71, 1.98) 2 losses: OR 0.64 95% CI (0.54, 7.07) 3 losses: OR 3.51 95% CI (1.08, 11.35) 3: 1 loss: OR 0.97 95% CI (1.42, 0.66) 2 losses: OR 0.56 95% CI (1.23, 0.27) 3 losses: OR 2.37 95% CI (5.7, 0.99) |

(*Continued*)

**Table 1.** (Continued)

| Lead author and publication date (Country) | Study design | Duration of follow-up | Exposure | Gestation | Type of control | Outcome | Number of exposed with outcome | Number of controls with outcome | Result (OR, HR, RR, IRR, BDI, LCU, O/E) |
|---|---|---|---|---|---|---|---|---|---|
| *Mikkelsen and colleagues (2019), Denmark* [59] | Case-control | 20 years | At least 1 miscarriage | Not defined | No history of miscarriage | **1:** Breast cancer<br>**2:** Ovarian cancer<br>**3:** Endometrial cancer<br>**4:** Cervical cancer | **1:**<br>1 loss: 1,926<br>2 losses: 424<br>3 losses: 157<br>**2:**<br>1 loss: 113<br>2 losses: 32<br>3 losses: 10<br>**3:**<br>1 loss: 125<br>2 losses: 22<br>3 losses: 7<br>**4:**<br>1 loss: 195<br>2 losses: 35<br>3 losses: 15 | **1:** 8,411<br>**2:** 581<br>**3:** 580<br>**4:** 878 | **1:**<br>1 loss: OR 0.98 95% CI (0.93, 1.04)<br>2 losses: OR 0.99 95% CI (0.89, 1.1)<br>3 losses: OR 0.97 95% CI (0.81, 1.16)<br>**2:**<br>1 loss: OR 0.86 95% CI (0.69, 1.07)<br>2 losses: OR 1.18 95% CI (0.8, 1.72)<br>3 losses: OR 1.11 95% CI (0.56, 2.17)<br>**3:**<br>1 loss: OR 0.93 95% CI (0.8, 1.11)<br>2 losses: OR 0.69 95% CI (0.44, 1.09)<br>3 losses: OR 0.75 95% CI (0.34, 1.66)<br>**4:**<br>1 loss: OR 0.95 95% CI (0.8, 1.11)<br>2 losses: OR 0.78 95% CI (0.55, 1.12)<br>3 losses: OR 0.9 95% CI (1.54, 0.53) |
| | | | | | | **5:** Bladder cancer<br>**6:** Renal cancer<br>**7:** Lung cancer<br>**8:** GI cancer<br>**9:** Brain cancer<br>**10:** Melanoma | **5:**<br>1 loss: 24<br>2 losses: 7<br>**6:**<br>1 loss: 76<br>2 losses: 19<br>**7:**<br>1 loss: 341<br>2 losses: 64<br>**8:**<br>1 loss: 482<br>2 losses: 105<br>3 losses: 38<br>**9:**<br>1 loss: 58<br>2 losses: 12<br>3 losses: 9<br>**10:**<br>1 loss: 538<br>2 losses: 87<br>3 losses: 30 | **5:** 124<br>**6:** 8<br>**7:** 28<br>**8:** 2,149<br>**9:** 280<br>**10:** 2,360 | **5:**<br>1 loss: OR 0.83 95% CI (0.52, 1.32)<br>2 losses: OR 1.24 95% CI (0.54, 2.81)<br>**6:**<br>1 loss: OR 1.31 95% CI (0.99, 1.73)<br>2 losses: OR 95% CI 1.39 (0.83, 2.34)<br>**7:**<br>1 loss: OR 1.08 95% CI (0.95, 1.23)<br>2 losses: OR 0.91 95% CI (0.69, 1.19)<br>**8:**<br>1 loss: OR 0.95 95% CI (0.85, 1.05)<br>2 losses: OR 0.98 95% CI (1.2, 0.79)<br>3 losses: OR 1.06 95% CI (0.76, 1.53)<br>**9:**<br>1 loss: OR 0.93 95% CI (0.69, 1.26)<br>2 losses: OR 0.81 95% CI (0.44, 1.5)<br>3 losses: OR 2.09 95% CI (0.93, 4.12)<br>**10:**<br>1 loss: OR 0.96 95% CI (0.87, 1.06)<br>2 losses: OR 0.72 95% CI (0.57, 0.9)<br>3 losses: OR 0.69 95% CI (0.47, 1.01) |

*(Continued)*

**Table 1.** (Continued)

| Lead author and publication date (Country) | Study design | Duration of follow-up | Exposure | Gestation | Type of control | Outcome | Number of exposed with outcome | Number of controls with outcome | Result (OR, HR, RR, IRR, BDI, LCU, O/E) |
|---|---|---|---|---|---|---|---|---|---|
| | | | | | | **I1:** Haematological cancer **I2:** All cancers | **I1:**<br>1 loss: 213<br>2 losses: 49<br>3 losses: 22<br>**I2:**<br>Total: 5,828<br>1 loss: 4,506<br>2 losses: 963<br>3 losses: 359 | **I1:** 898<br>**I2:** 19,591 | **I1:**<br>1 loss: OR 1.01 95% CI (0.86, 1.19)<br>2 losses: OR 1.06 95% CI (0.78, 1.44)<br>3 losses: OR 1.43 95% CI (0.9, 2.25)<br>**I2:**<br>1 loss: OR 0.98 95% CI (0.95, 1.02)<br>2 losses: OR 0.98 95% CI (0.92, 1.05)<br>3 losses: OR 1.02 95% CI (0.91, 1.14) |
| Okoth and colleagues (2022), UK [21] | Retrospective cohort | 20.5 years | At least 1 miscarriage | Not defined | No history of miscarriage | **1:** HTN **2:** T2DM | 1: 1,995<br>2: 792 | 1: 6,637<br>2: 2,525 | **1:** IRR 1.07 95% CI (1.02, 1.12)<br>**2:** IRR 1.25 95% CI (1.15, 1.36) |
| Parrazzini and colleagues (1992), Italy [66] | Case-control | - | At least 1 miscarriage | Not defined | No history of miscarriage | Breast cancer | 631 | 2,406 | RR 0.9 95% CI (0.9, 1.1) |
| Pell and colleagues (2003), UK [24] | Retrospective cohort | 19 years | At least 1 miscarriage | Not defined | Singleton live birth with no history of loss | Cerebrovascular events: IS, intracranial haemorrhage, TIA | - | - | HR 1.49 95% CI (1.09, 2.03) |
| Peters and colleagues (2017), UK [18] | Retrospective cohort | 7 years | At least 1 miscarriage | Not defined | Singleton live birth with no history of loss | **1:** CHD **2:** Stroke **3:** All circulatory diseases | **1:**<br>1 loss: 1,220<br>2 losses: 304<br>3 losses: 121<br>**2:**<br>1 loss: 1,695<br>2 losses: 374<br>3 losses: 148<br>**3:**<br>1 loss: 3,390<br>2 losses: 779<br>3 losses: 281 | **1:**<br>12,629<br>**2:**<br>17,497<br>**3:**<br>39,089 | **1:**<br>1 loss: HR 1.04 95% CI (0.98, 1.1)<br>2 losses: HR 1.19 95% CI (1.06, 1.33)<br>3 losses: HR 1.26 95% CI (1.05, 1.51)<br>**2:**<br>1 loss: HR 1.04 95% CI (0.99, 1.09)<br>2 losses: HR 1.09 95% CI (0.99, 1.21)<br>3 losses: HR 1.15 95% CI (0.98, 1.36)<br>**3:**<br>1 loss: HR 1.01 95% CI (0.98, 1.05)<br>2 losses: HR 1.12 95% CI (1.05, 1.12)<br>3 losses: HR 1.12 95% CI (0.99, 1.26) |
| Peters and colleagues (2020), UK [10] | Retrospective cohort | 9.2 years | At least 1 miscarriage | Not defined | Singleton live birth with no history of loss | T2DM | 1 loss: 528<br>2 losses: 131 | 7,050 | 1 loss: HR 1.02 95% CI (0.94, 1.11)<br>2 losses: HR 1.06 95% CI (0.9, 1.26) |

(*Continued*)

**Table 1.** (Continued)

| Lead author and publication date (Country) | Study design | Duration of follow-up | Exposure | Gestation | Type of control | Outcome | Number of exposed with outcome | Number of controls with outcome | Result (OR, HR, RR, IRR, BDI, LCU, O/E) |
|---|---|---|---|---|---|---|---|---|---|
| Ranthe and colleagues (2013), Denmark [28] | Retrospective cohort | 40 years | At least 1 miscarriage | Not defined | Singleton live birth with no history of loss | 1: MI<br>2: Cerebral infarction<br>3: Renovascular HTN | **1:**<br>1 loss: 541<br>2 losses: 90<br>3 losses: 14<br>4 losses: 15<br>**2:**<br>1 loss: 650<br>2 losses: 131<br>3 losses: 33<br>4 losses: 19<br>**3:**<br>1 loss: 204<br>2 losses: 36<br>3 losses: 12<br>4 losses: 11 | **1:**<br>2,228<br>**2:**<br>3,220<br>**3:**<br>1,006 | **1:**<br>1 loss: Ratio of proportion 1.11 95% CI (1, 1.23)<br>2 losses: Ratio of proportion 1.18 95% CI (0.95, 1.45)<br>3 losses: Ratio of proportion 0.85 95% CI (0.5, 1.44)<br>4 losses: Ratio of proportion 2.08 95% CI (1.25, 3.45)<br>**2:**<br>1 loss: Ratio of proportion 1.13 95% CI (1.03, 1.23)<br>2 losses: Ratio of proportion 1.22 95% CI (1.02, 1.45)<br>3 losses: Ratio of proportion 1.43 95% CI (1.01, 2.01)<br>4 losses: Ratio of proportion 1.89 95% CI (1.2, 2.96)<br>**3:**<br>1 loss: Ratio of proportion 1.15 95% CI (0.99, 1.34)<br>2 losses: Ratio of proportion 1.12 95% CI (0.8, 1.56)<br>3 losses: Ratio of proportion 1.78 95% CI (1, 3.14)<br>4 losses: Ratio of proportion 3.78 95% CI (2.08, 6.85) |
| Reeves and colleagues (2006), UK [53] | Case-control | 6.6 years | At least 1 miscarriage | Not defined | No history of miscarriage | Breast cancer | 1 loss: 761<br>2 losses: 296 | 3,719 | 1 loss: RR 1.02 95% CI (0.94, 1.1)<br>2 losses: RR 1.2 95% CI (1.07, 1.35) |
| Rosenberg and colleagues (1988), USA [64] | Case-control | 8 years | At least 1 miscarriage | <24 weeks | No history of miscarriage | Breast cancer | 1 loss: 398<br>2 losses: 152<br>3 losses: 67 | 1,504 | 1 loss: RR 1 95% CI (0.8, 1.2)<br>2 losses: RR 0.9 95% CI (0.7, 1.1)<br>3 losses: RR 0.6 95% CI (0.4, 0.8) |
| Schwarzman and colleagues (2020), Israel [20] | Retrospective cohort | 26 years | At least 2 miscarriages | <20 weeks | No history of recurrent miscarriage | 1: Arterial thromboembolic events<br>2: Venous thromboembolic events | **1:**<br>Total: 19<br>**2:**<br>Total: 41 | **1:** 177<br>**2:** 328 | - |
| Smith and colleagues (2003), UK [24] | Retrospective cohort | 19 years | At least 1 miscarriage | <24 weeks | No history of miscarriage | Death from IHD | 1 loss: 48<br>3 losses: 4 | 177 | 1 loss: HR 1.48 95% CI (1.09, 2.02)<br>3 losses: HR 2.35 95% CI (0.87, 6.36) |

(*Continued*)

**Table 1.** (Continued)

| Lead author and publication date (Country) | Study design | Duration of follow-up | Exposure | Gestation | Type of control | Outcome | Number of exposed with outcome | Number of controls with outcome | Result (OR, HR, RR, IRR, BDI, LCU, O/E) |
|---|---|---|---|---|---|---|---|---|---|
| Toffol and colleagues (a) (2013), Finland [68] | Retrospective cross-sectional | - | At least 1 miscarriage | Not defined | No history of miscarriage | 1: Alcohol abuse<br>2: Alcohol dependence<br>3: Dysthymic disorder<br>4: Major depressive episode<br>5: Anxiety disorder | 1: 326<br>2: 324<br>3: 325<br>4: 326<br>5: 319 | 1: 1,071<br>2: 1,006<br>3: 1,019<br>4: 1,018<br>5: 995 | **1:**<br>Age 30–40: OR 1.524 95% CI (0.265, 8.776)<br>Age 41–50: OR 1.405 95% CI (0.136, 14.542)<br>**2:**<br>Age 30–40: OR 0.367 95% CI (0.105, 1.282)<br>Age 41–50: OR 1.967 95% CI (0.781, 4.954)<br>**3:**<br>Age 41–50: OR 1.405 95% CI (0.047, 1.268)<br>**4:**<br>Age 30–40: OR 1.063 95% CI (0.510, 2.214)<br>Age 41–50: OR 1.830 95% CI (1.005, 3.334)<br>**5:**<br>Age 30–40: OR 1.006 95% CI (0.398, 2.543)<br>Age 41–50: OR 0.772 95% CI (0.357, 1.670) |
| Toffol and colleagues (b) (2013), Finland [68] | Retrospective cross-sectional | 5 years | At least 1 miscarriage | Not defined | No history of miscarriage | 1: Depressive symptoms<br>2: Recent depression diagnosis<br>3: Recent psychiatric diagnosis<br>4: Anhedonia during the last year | 1: 461<br>2: 459<br>3: 459<br>4: 4,602 | 1: 1,684<br>2: 1,684<br>3: 1,684<br>4: 1,685 | **1:** Age 25–40: OR 0.710 95% CI (0.455, 1.109)<br>Age 41–50: OR 1.117 95% CI (0.722, 1.729)<br>**2:** Age 25–40: OR 95% CI 1.091 (0.521, 2.287)<br>Age 41–50: OR 0.824 95% CI (0.487, 2.050)<br>**3:** Age 25–40: OR 3.664 95% CI (0.865, 15.516)<br>Age 41–50: OR 0.824 95% CI (0.173, 3.932)<br>**4:** Age 25–40: OR 0.614 95% CI (0.377, 0.998)<br>Age 41–50: OR 1.328 95% CI (0.846, 2.085) |
| Wagner and colleagues (2015), the Netherlands [17] | Retrospective cohort | 62 years | At least 2 miscarriages | <24 weeks | Singleton live birth with no history of pregnancy loss | 1: IHD<br>2: Cerebrovascular disease<br>3: Composite cardiovascular outcomes | **1:**<br>1 loss: 272<br>2 losses: 30<br>3 losses: 7<br>**2:**<br>1 loss: 139<br>2 losses: 14<br>3 losses: 3<br>**3:**<br>1 loss: 1,207<br>2 losses: 126<br>3 losses: 23 | **1:** 1,440<br>**2:** 826<br>**3:** 6,841 | **1:** 1 loss: HR 0.9 95% CI (0.69, 1.15)<br>2 losses: HR 0.98 95% CI (0.71, 1.31)<br>**2:** 1 loss: HR 1.12 95% CI (0.77, 1.64)<br>2 losses: HR 1.27 95% CI (0.74, 3.41)<br>**3:** 1 loss: HR 1.19 95% CI (1.08, 1.31)<br>2 losses: HR 1.29 95% CI (1, 1.66)<br>3 losses: HR 1.32 95% CI (0.73, 2.4) |
| Winkelstein and colleagues (1958), USA [44] | Case-control | - | At least 1 miscarriage | 12 weeks | No history of miscarriage | MI | 25 | 63 | RR 1.70 95% CI (1.17, 2.48) |
| Winkelstein and Rekate (1964), USA [55] | Case-control | - | At least 1 miscarriage | 12 weeks | No history of miscarriage | Atherosclerotic heart disease | 55 | 183 | RR 0.76 95% CI (0.65, 0.89) |
| Xu and colleagues (2004), China [51] | Case-control | 3 years | At least 1 miscarriage | Not defined | No history of miscarriage | Endometrial cancer | 66 | 304 | OR 1.03 95% CI (0.67, 1.58) |

(Continued)

**Table 1.** (Continued)

| Lead author and publication date (Country) | Study design | Duration of follow-up | Exposure | Gestation | Type of control | Outcome | Number of exposed with outcome | Number of controls with outcome | Result (OR, HR, RR, IRR, BDI, LCU, O/E) |
|---|---|---|---|---|---|---|---|---|---|
| *Yamada and colleagues (2017), Japan* [12] | Prospective cohort | 18 years | At least 1 miscarriage | Not defined | No history of miscarriage | **1:** Death from stroke (total)<br>**2:** Death from IS<br>**3:** Death from HS<br>**4:** Death from intracranial haemorrhage<br>**5:** Subarachnoid haemorrhage<br>**6:** Death from CHD:<br>**7:** Total cardiovascular disease | **1:**<br>1 loss: 205<br>2 losses: 134<br>**2:**<br>1 loss: 65<br>2 losses: 41<br>**3:**<br>1 loss: 90<br>2 losses: 61<br>**4:**<br>1 loss: 45<br>2 losses: 26<br>**5:**<br>1 loss: 45<br>2 losses: 35<br>**6:**<br>1 loss: 94<br>2 losses: 56<br>**7:**<br>Total: 767 | **1:** 901<br>**2:** 302<br>**3:** 325<br>**4:** 181<br>**5:** 144<br>**6:** 388<br>**7:** 2,048 | **1:**<br>1 loss: HR 0.91 95% CI (0.78, 1.06)<br>2 losses: HR 0.84 95% CI (0.7, 1.02)<br>**2:**<br>1 loss: HR 0.96 95% CI (0.73, 1.26)<br>2 losses: HR 0.95 95% CI (0.68, 1.33)<br>**3:**<br>1 loss: HR 0.95 95% CI (0.75, 1.21)<br>2 losses: HR 0.83 95% CI (0.63, 1.1)<br>**4:**<br>1 loss: HR 0.88 95% CI (0.63, 1.23)<br>2 losses: HR 0.68 95% CI (0.45, 1.03)<br>**5:**<br>1 loss: HR 1.04 95% CI (0.74, 1.45)<br>2 losses: HR 1.01 95% CI (0.69, 1.47)<br>**6:**<br>1 loss: HR 1 95% CI (0.79, 1.25)<br>2 losses: HR 0.84 95% CI (0.63, 1.12)<br>**7:** HR 0.93 95% CI (0.84, 1.03) |

BDI, Beck Depression Inventory; CHD, coronary heart disease; CI, confidence interval; CVD, cardiovascular disease; GI, gastrointestinal; HC, hypercholesterolaemia; HR, hazard ratio; HS, haemorrhagic stroke; HTN, hypertension; IHD, ishaemic heart disease; IRR, incidence rate ratio; IS, ischaemic stroke; LCU, life change units; MI, myocardial infarction; OCD, obsessive compulsive disorder; O/E, observed to exposed ratio; OR, odds ratio; RR, relative risk; T2DM, type 2 diabetes mellitus; TIA, transient ischaemic attack.

**Table 2. Characteristics of all studies examining the effect of stillbirth in long-term maternal morbidity and mortality included in the systematic review.**

| Lead author and publication date (Country) | Study design | Follow-up (years) | Exposure | Gestation | Type of control | Outcome | Number of exposed with outcome | Number of controls with outcome | Result (OR, HR, RR, 95% CI) |
|---|---|---|---|---|---|---|---|---|---|
| Auger and colleagues (2021), Canada [35] | Prospective cohort | 29 | History of stillbirth | Loss >20 weeks | Pregnancy ending in live birth | Mortality | <5 | 6,965 | HR 1.68 95% CI (1.17, 2.41) |
| Barret and colleagues (2020), Sweden [9] | Prospective cohort | 20.7 | History of stillbirth | From 1973–2008: >28 weeks, 2008–2012: >22 weeks | Singleton live birth with no stillbirth | CKD and ESRF | 1: CKD: 202 2: ESRF: 34 | 1: CKD: 17,815 2: ESRF: 1,249 | 1: HR 1.26 95% CI (1.09, 1.45) 2: HR 2.25 95% CI (1.55, 3.25) |
| Bourne and colleagues (1968), UK [15] | Retrospective cohort | 2 | History of stillbirth | - | Singleton live birth with no stillbirth | Any type of psychological symptoms | 14 | 11 | No significant association |
| Boyle and colleagues (1996), Australia [36] | Prospective cohort | 30 | History of stillbirth | - | Singleton live birth with no stillbirth | Delusion symptoms | 6 | 5 | RR 2.78 95% CI (0.83, 9.38) |
| Brinton and colleagues (1983), USA [61] | Case-control | - | History of stillbirth | Loss >16 weeks | Singleton live birth with no stillbirth | Breast cancer | 9 | 985 | RR 0.81 95% CI (0.3, 2.2) |
| Calderon-Margalit and colleagues (2007), Israel [40] | Prospective cohort | 36.5 | History of stillbirth | Loss >28 weeks | No history of stillbirth | 1: CHD 2: Circulatory disease 3: Mortality from cancers | 9 | 180 | 1: HR 2.00 95% CI (1.02, 3.93) 2: HR 1.7 95% CI (1.02, 2.84) 3: HR 1.29 95% CI (0.78, 2.15) |
| Calle and colleagues (1995), USA [46] | Case-control | 7 | History of stillbirth | Loss >20 weeks | No history of stillbirth | Breast cancer | - | - | RR 1.06 95% CI (0.84, 1.34) |
| Coleman and colleagues (2013), Denmark [27] | Retrospective cohort | 25 | History of stillbirth | - | No history of loss | All-causes mortality | 9,320 | 246,252 | OR 2.815 95% CI (2.207, 3.592) |
| Egerup and colleagues (2020), Denmark [60] | Case-control | 40 | History of stillbirth | Loss >22 weeks | No history of loss | T2DM | 270 | 1,534 | OR 1.94 95% CI (1.69, 2.21) |
| Gallagher and colleagues (2011), USA [41] | Prospective cohort | 11 | History of stillbirth | - | No history of stillbirth | 1: IHD 2: IS 3: HS | 1: 1 loss: 13 2 losses: 4 2: 1 loss: 21 2 losses: 8 3: 1 loss: 21 2 losses: 8 | 1: 408 2: 482 3: 1,517 | 1: 1 loss: HR 0.76 95% CI (0.43, 1.31) 2 losses: HR 1.02 95% CI (0.38, 2.73) 2: 1 loss: HR 0.86 95% CI (0.55, 1.33) 2 losses: HR 1.36 95% CI (0.68, 2.75) 3: 1 loss: HR 1.09 95% CI (0.86, 1.40) 2 losses: HR 0.80 95% CI (0.44, 1.44) |
| Gravesteen and colleagues (2012), Norway [57] | Case-control | 28 | History of stillbirth | Loss ≥23 weeks | Singleton live birth with no stillbirth | Depression | 18 | 41 | OR 0.77 95% CI (0.37, 1.57) |
| Hogue and colleagues (2015), USA [58] | Case-control | 3 | History of stillbirth | Loss >20 weeks | Singleton live birth with no stillbirth | Depression | 270 | 461 | OR 2.00 95% CI (1.00, 3.80) |

*(Continued)*

**Table 2.** (Continued)

| Lead author and publication date (Country) | Study design | Follow-up (years) | Exposure | Gestation | Type of control | Outcome | Number of exposed with outcome | Number of controls with outcomes | Result (OR, HR, RR, 95% CI) |
|---|---|---|---|---|---|---|---|---|---|
| Horn and colleagues (2019), Norway [34] | Prospective cohort | 20 | History of stillbirth | Loss >20 weeks | Singleton live birth with no stillbirth | **1:** HTN<br>**2:** HC<br>**3:** T2DM | **1:**<br>Total: 511<br>1 loss: 470<br>2 losses: 41<br>**2:**<br>Total: 717<br>1 loss: 668<br>2 losses: 49<br>**3:**<br>Total: 147<br>1 loss: 133<br>2 losses: 14 | **1:** 16,744<br>**2:** 24,541<br>**3:** 3,229 | **1:**<br>Total: HR 1.08 95% CI (0.99, 1.18)<br>1 loss: HR 1.06 95% CI (0.9, 1.16)<br>2 losses: HR 1.13 95% CI (0.99, 1.83)<br>**2:**<br>Total: HR 1.01 95% CI (0.94, 1.09)<br>1 loss: HR 1.01 95% CI (0.93, 1.09)<br>2 losses: HR 1.09 95% CI (0.82, 1.45)<br>**3:**<br>Total: HR 1.36 95% CI (1.15, 1.61)<br>1 loss: HR 1.34 95% CI (1.12, 1.60)<br>2 losses: HR 1.59 95% CI (0.94, 2.71) |
| Hvidtjørn and colleagues (2015), Denmark [32] | Prospective cohort | 29 | History of stillbirth | From 1980–2004: >28 weeks, 2004–2008: >22 weeks | No history of stillbirth | **1:** Circulatory mortality<br>**2:** Cancer mortality<br>**3:** Renal mortality | **1:** 22<br>**2:** 34<br>**3:** 0 | **1:** 849<br>**2:** 3,210<br>**3:** 28 | **1:** HR 2.68 95% CI (1.72, 4.17)<br>**2:** HR 1.14 95% CI (0.80, 1.63)<br>**3:** HR N/A |
| Kharazmi and colleagues (2011), Germany [42] | Prospective cohort | 10.8 | History of stillbirth | - | No history of stillbirth | **1:** MI<br>**2:** Stroke | **1:** 7<br>**2:** 5 | **1:** 72<br>**2:** 102 | **1:** HR 3.43 95% CI (1.53, 7.72)<br>**2:** HR 1.81 95% CI (0.65, 5.05) |
| Kharazmi and colleagues (2012), Germany [30] | Prospective cohort | 15.2 | History of stillbirth | - | No history of stillbirth | T2DM | - | - | HR 1.20 95% CI (0.59, 2.44) |
| Maino and colleagues (2016), the Netherlands [8] | Case-control | 3 | History of stillbirth | Loss >22 weeks | No history of stillbirth | **1:** MI<br>**2:** IS<br>**3:** Any arterial thrombosis | **1:** 13<br>**2:** 13<br>**3:** 26 | **1:** 155<br>**2:** 106<br>**3:** 261 | **1:** OR 1.04 95% CI (0.39, 2.76)<br>**2:** OR 2.06 95% CI (0.81, 5.23)<br>**3:** OR 1.68 95% CI (0.79, 3.55) |
| Mikklesen and colleagues (2019), Denmark [59] | Case-control | 20 | History of stillbirth | - | No history of stillbirth | **1:** Breast cancer<br>**2:** Ovarian cancer<br>**3:** Endometrial cancer<br>**4:** Cervical cancer<br>**5:** Bladder cancer<br>**6:** Renal cancer<br>**7:** Lung cancer<br>**8:** GI cancer<br>**9:** Brain cancer<br>**10:** Haematological cancer<br>**11:** Melanoma<br>**12:** All cancers | **1:** 79<br>**2:** 9<br>**3:** 7<br>**4:** 16<br>**5:** <5<br>**6:** 5<br>**7:** 25<br>**8:** 28<br>**9:** <5<br>**10:** 22<br>**11:** 26<br>**12:** 230 | **1:** 1,010<br>**2:** 82<br>**3:** 79<br>**4:** 121<br>**5:** 14<br>**6:** 37<br>**7:** 176<br>**8:** 256<br>**9:** 27<br>**10:** 157<br>**11:** 275<br>**12:** 2,397 | **1:** OR 0.84 95% CI (0.66, 1.07)<br>**2:** OR 1.27 95% CI (0.60, 2.69)<br>**3:** OR 1.03 95% CI (0.43, 2.45)<br>**4:** OR 1.26 95% CI (0.74, 2.14)<br>**5:** OR N/A<br>**6:** OR 1.32 95% CI (0.51, 3.41)<br>**7:** OR 1.32 95% CI (0.86, 2.04)<br>**8:** OR 1.13 95% CI (0.76, 1.68)<br>**9:** OR N/A<br>**10:** OR 0.63 95% CI (0.29, 1.63)<br>**11:** OR 0.92 95% CI (0.61, 1.38)<br>**12:** OR 0.94 95% CI (0.83, 1.09) |

(Continued)

**Table 2.** (Continued)

| Lead author and publication date (Country) | Study design | Follow-up (years) | Exposure | Gestation | Type of control | Outcome | Number of exposed with outcome | Number of controls with outcome | Result (OR, HR, RR, 95% CI) |
|---|---|---|---|---|---|---|---|---|---|
| Pariente and colleagues (2014), Israel [16] | Retrospective cohort | 25 | History of stillbirth | Loss >24 weeks | No history of stillbirth | 1: CNP<br>2: CIP<br>3: SCE<br>4: CCE<br>5: TCH<br>6: Renal morbidity | 1: 21<br>2: 11<br>3: 41<br>4: 56<br>5: 92<br>6: 8 | 1: 974<br>2: 390<br>3: 1,266<br>4: 1,656<br>5: 3,117<br>6: 97 | 1: OR 1.06 95% CI (0.60, 1.60)<br>2: OR 1.40 95% CI (0.80, 2.70)<br>3: OR 1.70 95% CI (1.20, 2.20)<br>4: OR 1.80 95% CI (1.30, 2.30)<br>5: OR 1.50 95% CI (1.20, 1.80)<br>6: OR 3.10 95% CI (1.40, 6.6) |
| Parker and colleagues (2014) USA [31] | Prospective cohort | 10.6 | History of stillbirth | - | No history of stillbirth | 1: MI<br>2: IS | 1: 179<br>2: 97 | 1: 2,809<br>2: 1,406 | 1: OR 1.27 95% CI (1.07, 1.51)<br>2: OR 1.13 95% CI (0.89–1.43) |
| Peters and colleagues (2017), China [18] | Retrospective cohort | 7 | History of stillbirth | - | Singleton live birth with no stillbirth | 1: CHD<br>2: Stroke<br>3: Circulatory disease | **1:**<br>Total: 1,196<br>1 loss: 822<br>2 losses: 240<br>3 losses: 134<br>**2:**<br>Total: 2,138<br>1 loss: 1,355<br>2 losses: 368<br>3 losses: 145<br>**3:**<br>Total: 3,487<br>1 loss: 2,538<br>2 losses: 662<br>3 losses: 287 | 1: 13,078<br>2: 17,846<br>3: 40,052 | **1:**<br>Total: HR 1.00 95% CI (0.94, 1.07)<br>1 loss: HR 0.96 95% CI (0.90, 1.02)<br>2 losses: HR 95% CI 1.08 (0.95, 1.23)<br>3 losses: HR 95% CI 1.31 (1.10, 1.56)<br>**2:**<br>Total: HR 1.06 95% CI (1.01, 1.12)<br>1 loss: HR 1.05 95% CI (1.00, 1.11)<br>2 losses: HR 1.13 95% CI (1.02, 1.25)<br>3 losses: HR 0.99 95% CI (0.84, 1.17)<br>**3:**<br>Total: HR 1.07 95% CI (1.03, 1.11)<br>1 loss: HR 1.05 95% CI (0.98, 1.05)<br>2 losses: HR 1.12 95% CI (1.04, 1.21)<br>3 losses: HR 1.13 95% CI (1.01, 1.28) |
| Peters and colleagues (2019), China [18] | Retrospective cohort | 9.2 | History of stillbirth | - | Singleton live birth with no stillbirth | T2DM | Total: 560<br>1 loss: 432<br>2 losses: 128 | 7,149 | Total: HR 1.10 95% CI (0.99, 1.19)<br>1 loss: HR 1.10 95% CI (1.00, 1.20)<br>2 losses: HR 1.08 95% CI (0.90, 1.29) |
| Rådestad and colleagues (1996), Sweden [23] | Retrospective cohort | 3 | History of stillbirth | Loss >28 weeks | Singleton live birth with no stillbirth | Anxiety | 31 | 15 | RR 2.10 95% CI (1.20, 3.90) |
| Ranthe and colleagues (2013), Denmark [28] | Retrospective cohort | 40 | History of stillbirth | - | Singleton live birth with no stillbirth | 1: MI<br>2: IS<br>3: Renal HTN | 1: 56<br>2: 52<br>3: 22 | 1: 2,742<br>2: 4,001<br>3: 1,247 | 1: IRR 2.69 95% CI (2.06, 3.50)<br>2: IRR 1.74 95% CI (1.32, 2.28)<br>3: IRR 2.42 95% CI (1.59, 3.69) |
| Rao and colleagues (1994), India [45] | Case-control | - | History of stillbirth | - | No history of stillbirth | Breast cancer | 10 | 653 | RR 0.90 95% CI (0.60, 1.40) |
| Vance and colleagues (1991), Australia [22] | Retrospective cohort | 2.5 | History of stillbirth | Loss >20 weeks | Singleton live birth with no stillbirth | 1: Anxiety<br>2: Depression | 1: 28<br>2: 19 | 1: 17<br>2: 7 | 1: OR 3.90 95% CI (2.10, 10.50)<br>2: OR 6.90 95% CI (2.10, 22.5) |

*(Continued)*

**Table 2.** (Continued)

| Lead author and publication date (Country) | Study design | Follow-up (years) | Exposure | Gestation | Type of control | Outcome | Number of exposed with outcome | Number of controls with outcomes | Result (OR, HR, RR, 95% CI) |
|---|---|---|---|---|---|---|---|---|---|
| Winkelstein and colleagues (1958), USA [44] | Case-control | - | History of stillbirth | - | No history of stillbirth | MI | 13 | 6 | Not provided |
| Winkelstein and colleagues (1964), USA [55] | Case-control | - | History of stillbirth | - | No history of stillbirth | IHD | 17 | 42 | Not provided |
| Xu and colleagues (2004), China [51] | Case-control | 3 | History of stillbirth | - | No history of stillbirth | Endometrial cancer | 10 | 760 | OR 0.64 95% CI (0.27, 1.48) |

CCE, complex cardiovascular events; CHD, coronary heart disease; CI, confidence interval; CIP, cardiac invasive procedures; CKD, chronic kidney disease; CNP, cardiac non-invasive events; ESRF, end-stage renal failure; GI, gastrointestinal; HC, hypercholesterolaemia; HR, hazard ratio; HS, haemorrhagic stroke; HTN, hypertension; IHD, ischaemic heart disease; IS, ischaemic stroke; MI, myocardial infarction; OR, odds ratio; RR, relative risk; SCE, simple cardiovascular event; T2DM, type 2 diabetes mellitus; TCH, total cardiac hospitalizations.

resulted in a pooled estimate of 1.13 (95% CI [0.89, 1.44]; $p$ = 0.32, $I^2$ = 71%, 95% PI [0.26, 7.72]) (Fig K in S1 Appendix).

Nine studies looked at history of stillbirth and future maternal IHD [8,18,28,31,40–42,44,55]. Our analysis included 36,213 women with a history of stillbirth and 1,442,348 without a history, of which 1,507 (4.16%) of the former and 19,492 (1.35%) of the latter developed IHD. Our pooled RR demonstrated a greater risk of developing IHD after stillbirth (RR 1.56, 95% CI [1.30, 1.88]; $p$ < 0.001, $I^2$ = 75%, 95% PI [0.49, 5.15]) (Fig 2b).

**Cerebrovascular disease.** There were 8 studies which examined the likelihood of developing long-term cerebrovascular disease including haemorrhagic and ischaemic stroke after miscarriage [8,12,17,18,25,28,41,42]. Our meta-analysis included data from 7 studies involving 282,975 women with a history of miscarriage compared with 1,411,788 controls without a history. Of these 3,916 (1.38%) with a history of miscarriage and 24,147 (1.71%) without a history of miscarriage developed cerebrovascular disease. Our analysis yielded a pooled RR of 1.01 (95% CI [0.85, 1.21]; $p$ = 0.89, $I^2$ = 94%, 95% PI [0.74, 1.32]) (Fig 2c). The study by Pell and colleagues (2004) (miscarriage: 10,850; controls: 108,818) was excluded from our analysis as the data were unavailable. Subgroup analysis showed that women with a history of recurrent

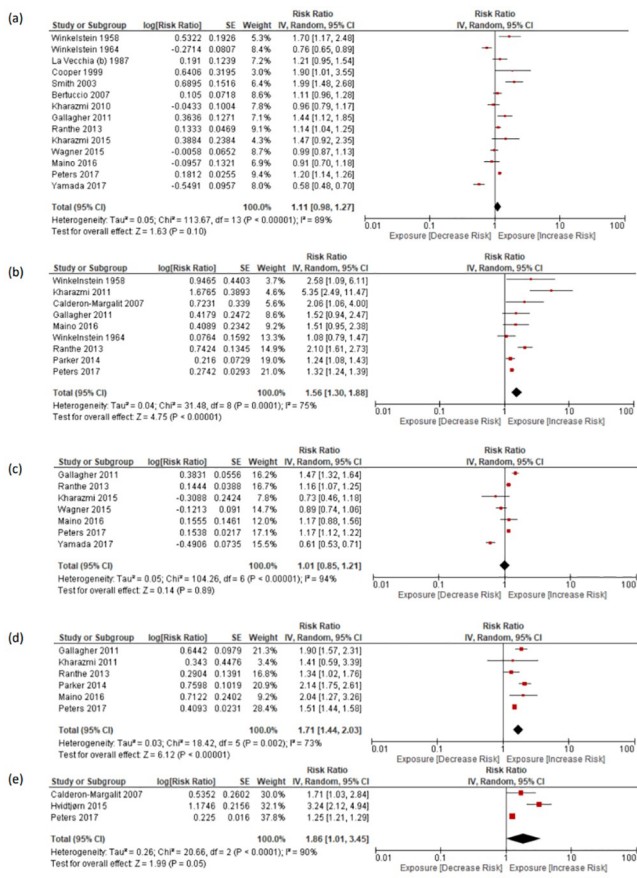

**Fig 2. Forest plot with pooled RR and 95% CIs for cardiovascular morbidity outcomes in women with and without history of miscarriage or stillbirth.** (a) IHD in women with miscarriage vs. no miscarriage, (b) IHD in women with stillbirth vs. no history of stillbirth, (c) cerebrovascular disease in women with miscarriage vs. no miscarriage, (d) cerebrovascular in women with stillbirths vs. no stillbirth, (e) overall circulatory disease in women with stillbirth vs. no stillbirth. CI, confidence interval; IHD, ischaemic heart disease; RR, risk ratio.

miscarriage did not have a higher risk of stroke, when compared to either women without a loss or women with a single loss (Figs N(b) and M(b) in S1 Appendix).

There were 6 studies assessing the risk of stillbirth with long-term cerebrovascular maternal morbidity, with the majority using ischaemic stroke as an outcome of interest [8,18,28,31,41,42]. These studies included a total of 35,539 women with a previous history of stillbirth and 1,418,143 controls, of which 2,143 (6.03%) and 25,618 (1.81%) had a stroke, respectively. The pooled RR was estimated at 1.71 (95% CI [1.44, 2.03]; $p < 0.001$, $I^2 = 73\%$, 95% PI [1.19, 2.42]) (Fig 2d).

**Circulatory/cardiovascular disease.** In the studies assessing overall cardiovascular risk, there were several studies which combined ischaemic stroke, haemorrhagic stroke, and IHD as a single outcome, defined as circulatory disease. There were 3 studies in women experiencing miscarriage [12,17,18] and 3 in women experiencing stillbirth [18,32,40]. There were 59,872 women with a history of miscarriage, compared to 342,095 controls without a history of miscarriage, and 26,075 women with a history of stillbirth compared to 1,126,946 controls without a history. Of these 6,573 (10.98%) women and 47,978 (14.02%) controls in the miscarriage arm and 3,524 (13.51%) women and 41,263 (3.66%) controls in the stillbirth arm developed circulatory disease. Meta-analysis was not conducted for the studies including women with miscarriage as they were considerably different and could not be grouped together.

The pooled RR for circulatory/cardiovascular disease was 1.86 (95% CI [1.01, 3.45]; $p = 0.05$, $I^2 = 90\%$, 95% PI [0.74, 4.10]) (Fig 2e) in women experiencing stillbirth.

**T2DM.** The association of miscarriage with the incidence of future T2DM was examined in 5 studies [10,21,30,34,60], and data were extracted from 3 studies [10,21,60], which included 154,435 women (events: 5,702 [3.69%]) with a history of miscarriage and 593,750 controls (events: 22,188 [3.74%]) without a history with a pooled RR of 0.82 (95% CI [0.43, 1.59]; $p = 0.57$, $I^2 = 100\%$, 95% PI [0.54, 1.51]) (Fig 3a).

Three studies [10,30,34] assessed the relationship of stillbirth and long-term T2DM, with 18,310 women experiencing stillbirth compared to 313,737 controls, of which 628 (3.43%) and 10,135 (3.23%) women developed T2DM respectively with a pooled RR of 1.16 (95% CI [1.07, 1.26]; $p < 0.001$, $I^2 = 0\%$, 95% PI [1.05, 1.35]) (Fig 3b).

**Renal disease.** There was only 1 study [28] investigating the risk of renal disease following miscarriage; therefore, meta-analysis was not possible. There were 5 studies investigating the risk of renal disease following stillbirth which included women experiencing renal cancer or other renal-associated mortality [9,16,28,32,59]. We pooled data only from studies where end-stage renal disease was defined as the outcome [9,16,28]. This included 3 studies with a total of 23,102 (events: 265 [1.15%]) exposed women and 2,868,042 (events: 20,408 [0.71%]) controls, resulting in a pooled RR of 1.97 (95% CI [1.51, 2.57]; $p < 0.001$, $I^2 = 39\%$, 95% PI [1.06, 4.72]) (Fig 3c).

**Malignant disease.** There were 15 studies assessed the association between miscarriage and future breast cancer diagnosis or mortality [19,33,46,47,49,50,52,53,59,61,62,64–67]. Data were extracted from all studies but the study by Goldacre and colleagues provided no extractable data and included 229,606 women with a history of miscarriage and 753,236 without a history, of which 6,381 (2.78%) and 23,514 (3.12%) developed breast cancer, respectively. The pooled RR was 0.99 (95% CI [0.93, 1.05]; $p = 0.69$, $I^2 = 79\%$, 95% PI [0.67, 1.48]) (Fig 4a). In a subgroup analysis, recurrent exposure to miscarriage did not demonstrate any difference in the risk of breast cancer when compared to either single or no history of loss (Figs K(b) and L (b) in S1 Appendix).

Four studies assessed the risk of miscarriage and the future diagnosis of ovarian cancer. They included a total of 65,898 women with a history of at least 1 miscarriage and 285,613 without, of which 908 (1.38%) in the former group and 2,697 (0.94%) in the latter developed

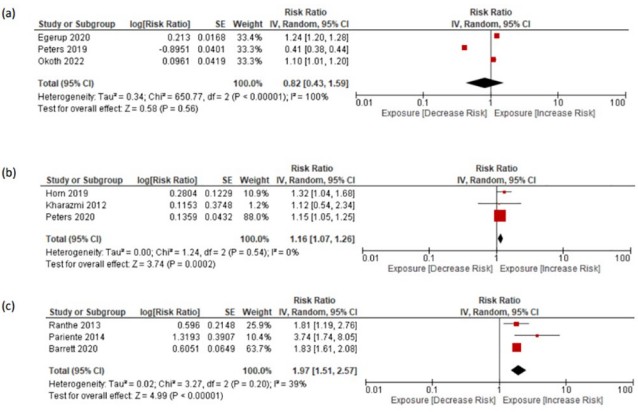

**Fig 3. Forest plots with pooled RR and 95% CIs for outcomes in women with and without history of miscarriage or stillbirth.** (a) T2DM in women with miscarriage vs. no miscarriage, (b) T2DM in women with stillbirth vs. no stillbirth, and (c) renal disease in women with stillbirth vs. no stillbirth. CI, confidence interval; RR, risk ratio; T2DM, type 2 diabetes mellitus.

ovarian cancer [19,43,56,59]. The pooled RR was 1.00 (95% CI [0.94, 1.06]; $p$ = 0.98, $I^2$ = 0%, 95% PI [0.78, 1.57]) (Fig 4c). Subgroup analysis comparing recurrent with a single miscarriage yielded similar results (Fig N(c) in S1 Appendix).

Two studies assessed the association of miscarriage with endometrial cancer [51,59] and 2 with cervical cancer [19,59]. These studies included a total of 12,240 women with a history of miscarriage and 116,425 controls, of which 498 (4.07%) in the exposed and 2,075 (1.78%) in the control arms developed uterine malignancies. The association of miscarriage with the incidence of uterine cancer yielded a pooled RR of 1.05 (95% CI [0.85, 1.31]; $p$ = 0.65, $I^2$ = 75%, 95% PI [0.83, 1.54]) (Fig 4d).

Four studies examined the risk of breast cancer in women who had experienced stillbirth. These studies comprised of a total of 1,129 women with a history of stillbirth and 118,412 without, with 98 (8.68%) of exposed and 12,306 (10.39%) controls developing breast cancer [45,46,59,61]. The pooled risk of the included studies demonstrated a small but significant reduction in the risk of breast cancer (RR: 0.80, 95% CI [0.67, 0.96]; $p$ = 0.02, $I^2$ = 0%, 95% PI [0.72, 0.93]) (Fig 4b). Moreover, endometrial [51,59] and ovarian [59] malignancy risk was examined in 2 and 1 studies, respectively, while cervical malignancy risk was assessed in 1 paper [59]. These studies investigated a total of 2,654 women with stillbirth and 276,553 without, of which 42 (1.58%) and 1,052 (0.38%) developed female malignancies, respectively. No meta-analysis was performed as not enough studies were retrieved.

Three studies assessed the risk of all cancer-related mortality in women with a previous history of stillbirth ($n$ = 8,609; events: 81) [32,40,59] compared to those without such loss (controls: $n$ = 885,036; events: $n$ = 6,245). Analysis showed an RR of 1.17 (97% CI [0.95, 1.45]; $p$ = 0.15, $I^2$ = 0%, 95% PI [1.13, 1.29]) (Fig 4e).

**Mental health illness.** Future risk of long-term depression, anxiety, stress, and alcohol dependence were assessed after pregnancy loss in 5 studies of women with a history of miscarriage [37–39,68,69] and in 6 studies in women with a history of stillbirth [15,22,23,36,57,58]. The paper by Toffol and colleagues included data from 2 separate studies, and such we considered them as different cohorts, as reflected by the annotation in Table 1. These studies included 1,089 women with pregnancy loss and were compared with 4,515 controls, with 89 (8.20%) of the former and 215 (4.76%) of the latter developing long-term mental health

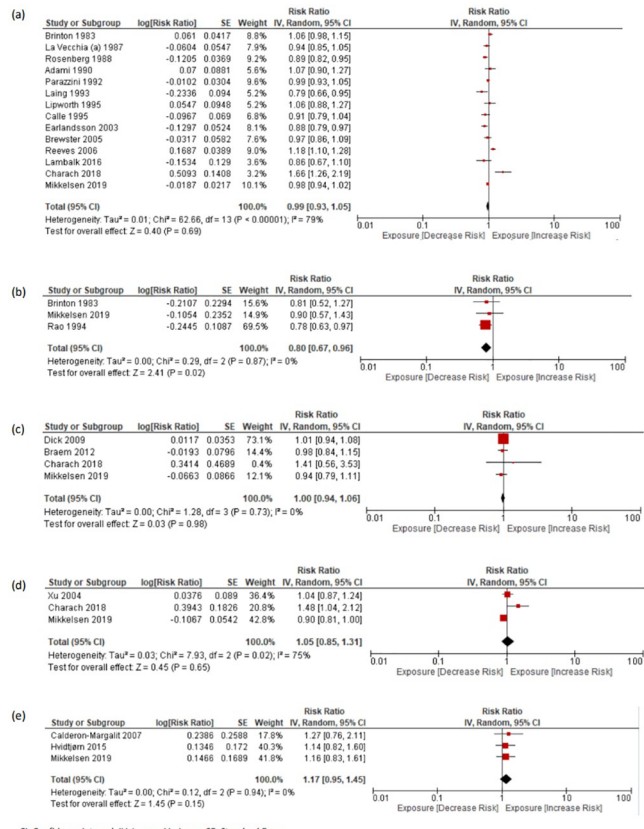

**Fig 4. Forest plots with pooled RR and 95% CIs for oncological morbidity outcomes in women with and without history of miscarriage or stillbirth.** (a) Breast cancer in women with miscarriage vs. no miscarriage, (b) breast cancer in women with stillbirth vs. no stillbirth, (c) ovarian cancer in women with miscarriage vs. no miscarriage, (d) uterine malignancies in women miscarriage vs. no miscarriage, (e) all cancer-related mortality in women with stillbirth vs. no stillbirth. CI, confidence interval; RR, risk ratio.

outcomes. The pooled RR of long-term depression following miscarriage was 1.78 (95% CI [0.88, 3.62]; $p = 0.11$, $I^2 = 88\%$, 95% PI [1.13, 4.16]) (Fig 5a). Four studies examined depression risk in women with history of stillbirth [22,36,57,58] including 545 (events: 256 [46.97%]) women with loss and 1,172 (events: 388 [33.12%]) controls, resulting in a pooled RR of 1.88

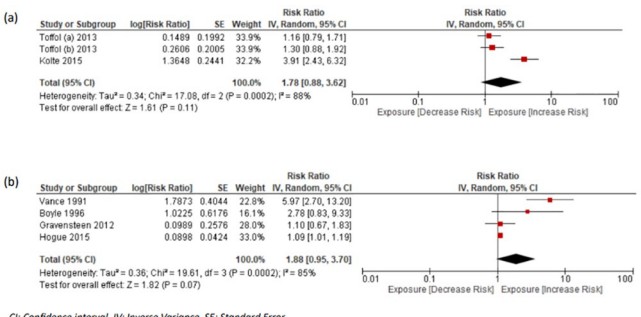

**Fig 5. Forest plots with pooled RR and 95% CIs for depression in women with and without history of (a) miscarriage or (b) stillbirth.** CI, confidence interval; RR, risk ratio.

(95% CI [0.95, 3.70]; $p$ = 0.07, $I^2$ = 85%, 95% PI [0.34, 9.51]) (Fig 5b). There were 2 studies exploring the risk of anxiety related to a history of stillbirth [22,23] and meta-analysis was not possible.

## Discussion

We conducted a systematic review and meta-analysis examining the association between fetal loss and long-term maternal morbidity or mortality in women experiencing either miscarriage or stillbirth.

Our results suggest that women experiencing stillbirth have an increased risk of developing IHD and cerebrovascular disease by 56% and 71%, respectively. We showed a 16% increase in the risk of T2DM after stillbirth. Additionally, women with a history of stillbirth are almost twice as likely to develop renal disease in the future. There was no evidence of an association between history of stillbirth, single or recurrent miscarriage with future psychiatric disease, ovarian or any other female malignancies.

Cardiovascular disease (CVD) is one of the most commonly examined long-term maternal morbidities after pregnancy loss [10,17,28,40]. Individual population-based cohort studies demonstrate an increase in the risk of IHD ranging from 1.7 [44] to 2.7 [28] times compared to women with no history of pregnancy loss. However, there are other studies not supporting such an association in this systematic review [8,41,54]. Our results are in keeping with published systematic reviews in this area suggesting a history of stillbirth increases the risk of CVD by up to 50% [70]. Our work provides an updated and comprehensive analysis, covering a longer follow-up period. However, we acknowledge that risk factors for stillbirth such as pre-eclampsia and gestational diabetes are established risk factors for CVD. Given the available data in aggregated studies, we were unable determine the impact of these factors individually in this review. It is possible that data on combined risk factors (such as gestational diabetes and preeclampsia combined, along with others) may have influenced our findings. Our results suggest that a history of stillbirth maybe an indicator of cardiovascular risk over and above established risk factors such as gestational diabetes for long-term morbidity and merits further research.

There are several studies examining the association of miscarriage with long-term development of CVD with conflicting results with some of these suggesting that miscarriage increases future CVD [24,28,44,48], whereas the majority were unable to demonstrate an effect [17,26,41]. A previous meta-analysis by Oliver-Williams and colleagues concluded that miscarriage increases the risk of CVD by 45%, while recurrent exposure makes it twice as likely [11]. Our results, which update these analyses, did not find a substantial relationship between miscarriage including recurrent loss with developing CVD. This may be due to the addition of contemporary studies with more complete longitudinal data and case definitions.

Cerebrovascular disease, while closely associated with IHD, is often examined separately because of the different pathological mechanisms and related morbidity. Studies investigating the association of history of stillbirth with both ischaemic and haemorrhagic strokes are varied, with some suggesting an increase in the risk [18,28], while others did not show an association [8,41,42]. Our analysis is the first pooled analysis of cerebrovascular morbidity to our knowledge in women experiencing a stillbirth. Our results suggest an overall higher risk of morbidity in women with a history of stillbirth, again suggesting a stillbirth maybe a marker of early cardiovascular disease.

Stillbirth and miscarriage are pregnancy losses occurring at different gestational ages and it has been hypothesised that they may share an underlying aetiology related to placental malformation, thrombotic and vascular changes. It is suggested that women experiencing stillbirth

are exposed to these changes for a longer period and therefore are further predisposed to long-term cerebrovascular disease. However, other risk factors such as hyperlipidaemia, hyperglycaemia, preeclampsia, insulin resistance, and gestational diabetes [71,72] contribute to both higher risk of stillbirth and subsequent greater risk of CVD or other metabolic consequences. Early miscarriages (either single or recurrent) are more likely to be multifactorial with vascular factors in only a small percentage of them which would dilute the impact on the overall CVD risk [73].

The metabolic disturbances and the related hyperinsulinemia and insulin resistance in pregnancy [74,75] can lead to the future development of T2DM in women with a history of stillbirth. Our results support this, as we demonstrated a 16% increase in the exposed group compared to controls. A prospective cohort study by Horn and colleagues suggests a 45% increase in the risk [34]. History of stillbirth can be on the causal pathway between pregnancy hyperglycaemia and T2DM; this may dilute the direct association between stillbirth and T2DM because of the strength of the association of other mediating and risk factors such as obesity, diet, and ethnicity [76,77].

Renal disease and hypertensive disorders following stillbirth have been explored in the literature. Specifically, data by Barret and colleagues illustrate a 30% increase in the risk of developing CKD, following stillbirth, as renovascular hypertension (RHTN) is common in these women [9]. Our pooled analyses support this. Whether this is mediated through hypertensive disorders in pregnancy is unclear, but our analysis shows the potential of identifying women at risk of future disease.

The correlation between history of stillbirth and occurrence of maternal malignancy is inconsistent in the literature. Breast, ovarian, endometrial, and cervical cancers are the most commonly reported neoplasms [45,46,59,61]. Our meta-analysis suggests that women experiencing a stillbirth are at a lower risk of developing breast cancer in the future. The mechanisms mediating the modest risk reduction for history of stillbirth warrant further laboratory, clinical, and epidemiological investigations but may be partially related with reduced gestational duration and therefore exposure to high levels of reproductive hormones. A well-defined comparator group and good phenotype of breast cancer types may shed more light on the mechanisms underlying this association. Our analysis involved a small number of studies, something necessitating further research with replication of results in future studies.

Pregnancy loss of any type is considered a significant psychological stressor with detrimental effects to maternal wellbeing. Women experiencing loss often exhibit signs of guilt, anger, sadness, and, in some cases, depression [5]. They often view stillbirth or miscarriage as the loss of a family member and therefore undergo a mourning process [5]. Studies indicate that the rate of depression and anxiety-related illness observed in mothers with a history of pregnancy loss is up to 55% [78–80] and 45% of exposed women, respectively [7,81,82]. Research suggests that signs of psychological morbidities remain evident up to 3 years after stillbirth, even with the birth of another child [80,83]. Our results demonstrate an association, albeit not significant, between history of stillbirth or miscarriage and depression beyond 1 year following the loss. The heterogeneity of the tools for assessing depression, in the duration of follow-ups and methodological approaches, may have weakened the association in our pooled analysis [23,36,58]. There is a need to further assess this relationship with more objective tools and a larger more representative population.

Our meta-analysis has several strengths. To our knowledge, this study is the first comprehensive systematic review evaluating the likelihood of future morbidity and mortality across all organ systems in women with a previous exposure to any type of pregnancy loss. Our search involved 6 global electronic databases with no time or language restrictions, including a search of the grey literature. Secondly, the majority of the included studies scored highly on the NOS

quality assessment tool, thereby indicating low risk of methodological bias in the aggregated data. Lastly, the heterogeneity reported in some of the outcomes including malignant and renal disease in the stillbirth cohort was small, indicating appropriate grouping of results.

However, there are also limitations. Firstly, the heterogeneity for the circulatory, metabolic, and psychiatric outcomes was relatively high for both the stillbirth and miscarriage groups. This could be explained by the small number of studies, which might not be fully representative of the prevalence of these conditions in the general population. It can be argued that while $I^2$ is able to compare heterogenicity of meta-analysis with different numbers of studies, it has been shown to also increase in value with increasing sample size. In our meta-analysis, some of the included studies involve many thousands of women and thus tau$^2$ figures, which is unaffected by sample size, are more useful in assessing heterogeneity [84]. Additionally, the follow-up period, sample size reported, as well as definitions of stillbirth and miscarriage also varied between studies, further contributing to this variation. Also, we note that while we used the most adjusted results from the studies, we recognise that there is some degree of confounding which cannot be accounted for. This is because the included studies are observational and thus, we extracted aggregated data with variable degrees of adjustment. This is specifically relevant to the association of stillbirth with CVD as we are unable to adjust for preeclampsia or gestational diabetes in the aggregated data. Some of the reported outcomes, for example, breast cancer in women with a history of stillbirth involved a small number of studies, something which warrants further research to investigate reliability of the results. Lastly, most of the retrieved studies investigated the population from high-income regions, suggesting our findings may not be generalisable in middle- and low-income settings.

Our results show women experiencing a fetal loss, and specifically stillbirth, are at a higher risk of IHD, cerebrovascular disease, T2DM, as well as renal morbidities. Although it is unclear whether the associations are causal, our results identify a group of high-risk women for subsequent disease that may benefit from close monitoring and primary prevention mechanisms for risk modification. Further research which will phenotype better this high-risk group and assess the additional value of stillbirth beyond established risk factors such as preeclampsia and gestational diabetes is warranted.

## Supporting information

**S1 Appendix. Supporting information.** Fig A in S1 Appendix: Key Word Strategy. Fig B in S1 Appendix: Quality assessment of cohort studies using Adapted Newcastle Ottawa Scale. Fig C in S1 Appendix: Quality assessment of case-controlled studies using Adapted Newcastle Ottawa Scale. Fig D in S1 Appendix: Quality assessment of cross-sectional studies using Adapted Newcastle Ottawa Scale. Fig E in S1 Appendix: Funnel plots of the association of miscarriage and (a) ischaemic heart disease and (b) breast cancer. Fig F in S1 Appendix: Forrest plots presenting pooled RRs for development of (a) ischaemic heart disease, (b) cerebrovascular disease and (c) circulatory diseases for the miscarriage arm using the Mantel–Haenszel random effects model with dichotomous data in meta-analysis. Fig G in S1 Appendix: Forrest plots presenting pooled RRs for development of (a) type 2 diabetes mellitus and (b) depression for the miscarriage arm using the Mantel–Haenszel random effects model with dichotomous data in meta-analysis. Fig H in S1 Appendix: Forrest plots presenting pooled RRs for development of (a) breast cancer, (b) ovarian cancer, and (c) uterine malignancies for the miscarriage arm using the Mantel–Haenszel random effects model with dichotomous data in meta-analysis. Fig I in S1 Appendix: Forrest plots presenting pooled RRs for development of (a) ischaemic heart disease, (b) cerebrovascular disease, (c) breast cancer, and (d) ovarian cancer for recurrent vs. single miscarriage using the Mantel–Haenszel random effects model with

dichotomous data in meta-analysis. Fig J in S1 Appendix: Forrest plots presenting pooled RRs for development of (a) ischaemic heart disease, (b) cerebrovascular disease, and (c) breast cancer for recurrent vs. no miscarriage using the Mantel–Haenszel random effects model with dichotomous data in meta-analysis. Fig K in S1 Appendix: Forrest plots presenting pooled RRs for development of (a) ischaemic heart disease, (b) cerebrovascular disease, (c) breast cancer, and (d) ovarian cancer for recurrent vs. single miscarriage using the Generic inverse variance random effects model meta-analysis. Fig L in S1 Appendix: Forrest plots presenting pooled RRs for development of (a) ischaemic heart disease, (b) cerebrovascular disease, and (c) breast cancer for recurrent vs. no miscarriage using the Generic inverse variance random effects model with dichotomous data in meta-analysis. Fig M in S1 Appendix: Forrest plots presenting pooled RRs for development of (a) ischaemic heart disease, (b) cerebrovascular disease, and (c) circulatory disease for stillbirth arm using the Mantel–Haenszel random effects model with dichotomous data in meta-analysis. Fig N in S1 Appendix: Forrest plots presenting pooled RRs for development of (a) breast cancer, (b) female malignancies, and (c) all malignancies for stillbirth arm using the Mantel–Haenszel random effects model with dichotomous data in meta-analysis. Fig O in S1 Appendix: Forrest plots presenting pooled RRs for development of (a) type 2 diabetes, (b) renal disease, (c) depression for stillbirth arm using the Mantel–Haenszel random effects model with dichotomous data in meta-analysis. Table A in S1 Appendix: Outcome definitions of the miscarriage arm of the meta-analysis. Table B in S1 Appendix: Outcome definitions of the stillbirth arm of the meta-analysis. Table C in S1 Appendix: Covariate/Confounding variables the researchers adjusted for in the studies of miscarriage. Table D in S1 Appendix: Covariate/Confounding variables the researchers adjusted for in the studies of stillbirth.
(DOCX)

**S1 PRISMA Checklist. PRISMA checklist.**
(DOCX)

## Author Contributions

**Conceptualization:** Jahnavi Daru, Stamatina Iliodromiti.

**Data curation:** Florentia Vlachou, Despoina Iakovou, Jahnavi Daru.

**Formal analysis:** Florentia Vlachou, Despoina Iakovou, Jahnavi Daru, Stamatina Iliodromiti.

**Investigation:** Jahnavi Daru.

**Methodology:** Jahnavi Daru, Stamatina Iliodromiti.

**Project administration:** Florentia Vlachou, Despoina Iakovou.

**Supervision:** Jahnavi Daru, Siobhan Quenby, Stamatina Iliodromiti.

**Visualization:** Florentia Vlachou.

**Writing – original draft:** Florentia Vlachou, Despoina Iakovou, Jahnavi Daru.

**Writing – review & editing:** Jahnavi Daru, Rehan Khan, Litha Pepas, Siobhan Quenby, Stamatina Iliodromiti.

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
