## [Editor Report · Decision Letter 0]

6 Mar 2023

Dear Dr Daru, 

Thank you for submitting your manuscript entitled "Fetal loss and long-term maternal morbidity and mortality; a systematic review and meta-analysis." for consideration by PLOS Medicine.

Your manuscript has now been evaluated by the PLOS Medicine editorial staff as well as by an academic editor with relevant expertise and I am writing to let you know that we would like to send your submission out for external peer review.

Please re-submit your manuscript within two working days, i.e. by Mar 08 2023 11:59PM.

Kind regards,

Callam Davidson

Senior Editor

PLOS Medicine

---

## [Decision Letter · Decision Letter 1]

17 May 2023

Dear Dr. Daru,

Thank you very much for submitting your manuscript "Fetal loss and long-term maternal morbidity and mortality; a systematic review and meta-analysis." (PMEDICINE-D-23-00438R1) for consideration at PLOS Medicine. 

Your paper was evaluated by an associate editor and discussed among all the editors here. It was also discussed with an academic editor with relevant expertise, and sent to independent reviewers, including a statistical reviewer. The reviews are appended at the bottom of this email and any accompanying reviewer attachments can be seen via the link below:

[LINK]

In light of these reviews, I am afraid that we will not be able to accept the manuscript for publication in the journal in its current form, but we would like to consider a revised version that addresses the reviewers' and editors' comments. Obviously we cannot make any decision about publication until we have seen the revised manuscript and your response, and we plan to seek re-review by one or more of the reviewers. 

We expect to receive your revised manuscript by Jun 07 2023 11:59PM. Please email us (plosmedicine@plos.org) if you have any questions or concerns.

We look forward to receiving your revised manuscript. 

Sincerely,

Alexandra Schaefer, PhD

PLOS Medicine

plosmedicine.org

ACADEMIC EDITOR COMMENTS

The reviewers raise a number of issues with this paper - the majority of which I agree with.

My main concerns are

The potential for confounding is very large and this was not addressed. Even if papers did appropriately adjust for potential confounding (which was not presented) the potential for residual confounding is high. It is well recognised that pregnancy complications and cardiovascular disease share similar risk factors.

The authors argue that even if the association is due to shared risk factors it is still useful to identify a high risk population so they can be followed up. I disagree. There are much stronger cardiovascular risk factors than stillbirth and well validated risk tools for assessment of cardiovascular risk, therefore the clinical benefit of using stillbirth to identify risk is questionable (especially when the evidence for the association is not strong).

The timing of exposure (pregnancy loss) and outcome (maternal health) is crucial in these analyses- ie did the pregnancy loss precede development of illness or did the pregnancy loss occur on the background of illness. The intervals between exposure and outcomes also needs to be considered. These were not addressed.

There was a lack of detail in the definitions of stillbirth and miscarriage. The lack of detail on gestation of loss, definitions, and cause or suspected cause of loss is problematic as different causes are likely to have different associations with maternal disease. The authors state “ Stillbirth and miscarriage are pregnancy losses occurring at different gestational ages and it has been hypothesised that they may share an underlying aetiology related to placental malformation, thrombotic and vascular changes.”.

This may be true for proportion of, mainly later miscarriages, and some stillbirths. However, early miscarriages are more likely to have other causes (for example chromosomal abnormality) and numerically these are by far the most common type of loss.

Absolute rates/risk reduction and numbers were not included. This makes interpretation difficult.

As the statistical reviewers additional analyses showed - there is enormous variation in rate of IHD in non-stillbirth arm. This may relate to follow up time or other differences - but it does raise questions on the rationale for combining these data. The other additional analyses by the reviewer suggest that the evidence base is not strong.

GENERAL

Please respond to all editor and reviewer comments detailed below in full.

Please cite the reference numbers in square brackets (e.g., “We used the techniques developed by our colleagues [19] to analyze the data”). Citations should be preceding punctuation.

Please cite your Supporting Information as outlined here: https://journals.plos.org/plosmedicine/s/supporting-information

Please include page numbers and line numbers in the manuscript file. Use continuous line numbers (do not restart the numbering on each page).

Please ensure to be consistent in the use of numbers as words or numerals.

Please adjust the number format using a consistent format (1000 or 1,000). 

Please check your manuscript carefully for grammar, spelling and punctuation.

ABSTRACT

Please report your abstract according to PRISMA for abstracts, following the PLOS Medicine abstract structure (Background, Methods and Findings, Conclusions) http://www.plosmedicine.org/article/info:doi/10.1371/journal.pmed.1001419 .

Please provide the types of study designs included, and synthesis/appraisal methods.

Abstract Methods and Findings:

* Please ensure that all numbers presented in the abstract are present and identical to numbers presented in the main manuscript text.

PLOS Medicine requests that main results are quantified with 95% CIs as well as p values. Please include. When reporting p values please report as p<0.001 and where higher as the exact p value p=0.002, for example. For the purposes of transparent data reporting, if not including the aforementioned please clearly state the reasons why not.

Please include any important dependent variables that are adjusted for in the analyses.

Throughout, suggest reporting statistical information as follows to improve clarity for the reader “22% (95% CI [13%,28%]; p</=)”. Please amend throughout the abstract and main manuscript.

Please note the use of commas to separate upper and lower bounds, as opposed to hyphens as these can be confused with reporting of negative values.

Please remove the “Key words” paragraph from the Abstract.

AUTHOR SUMMARY

At this stage, we ask that you include a short, non-technical Author Summary of your research to make findings accessible to a wide audience that includes both scientists and non-scientists. The Author Summary should immediately follow the Abstract in your revised manuscript. This text is subject to editorial change and should be distinct from the scientific abstract. Please see our author guidelines for more information: https://journals.plos.org/plosmedicine/s/revising-your-manuscript#loc-author-summary.

The summary should include 2-3 single sentence, individual bullet points under each of the questions.

It may be helpful to review currently published articles for examples which can be found on our website here https://journals.plos.org/plosmedicine/

INTRODUCTION

p.4: Please change “[…] at risk of developing of future cardiovascular, […]” to “[…] at risk of developing future cardiovascular, […]”.

METHODS AND RESULTS

Please report your SR/MA according to the PRISMA guidelines provided at the EQUATOR site.

http://www.equator-network.org/reporting-guidelines/prisma/

Please provide the completed PRISMA checklist. 

Please add the following statement, or similar, to the Methods: "This study is reported as per the Preferred Reporting Items for Systematic Reviews and Meta-Analyses (PRISMA) guideline (S1 Checklist)."Please update your PROSPERO entry (ID: CRD42021225370).

Please update your search to the present time.

Please evaluate evidence of publication bias.

PLOS Medicine requests that main results are quantified with 95% CIs as well as p values. Please include. When reporting p values please report as p<0.001 and where higher as the exact p value p=0.002, for example. For the purposes of transparent data reporting, if not including the aforementioned please clearly state the reasons why not.

Please include any important dependent variables that are adjusted for in the analyses.

Suggest reporting statistical information as detailed above – see under ABSTRACT

p. 5: “The following electronic databases were searched from inception to May 2022 to identify studies

exploring the effects of fetal loss on subsequent maternal morbidity and mortality” – Please include the name of the databases included in your study.

p.8: Please correct “The included 32 cohort studies […]”.

p.8.: Under “Study Characteristics” you write that of 25 case control studies two were retrospective cross-sectional studies. This would mean that your review included 57 studies in total. Please re-phrase the sentence if you want to describe the retrospective cross-sectional studies as a distinct set of studies. 

p.9: Please define “across Europe.”.

p.9: Due to repetition, please re-phrase “The mean maternal age at the time of fetal loss was on average 23 years at the time of loss (Pell et al., 2004) […]”. Editorial suggestion: “The mean maternal age was on average 23 years at the time of fetal loss (Pell et al., 2004) […]”.

p.9: Please write “Table 1”/“Figure 1” instead of “table 1”/”figure 1” and check throughout your manuscript.

p.9: “There were 29 studies looking at women experiencing a stillbirth with most being as low risk of bias, with only one older study identified as being at high risk of bias”. – change “as low risk” to “at low risk”. 

p.9: Please elaborate/clarify whether the bias identified could be attributed to the study 'vintage' or refer to “only one study...” instead of “only one older study”.

p.9: Please change “[…] went on to have IHD, […]” to “[…] developed IHD, […]”.

p.10: “Similarly, recurrent loss compared with a single loss resulted in a pooled estimate 1.13 (95%

CI: 0.89-1.44, I2 = 71%)”. – change “in a pooled estimate 1.13” to “in a pooled estimate of 1.13”.

p.11: “The pooled risk ratio […]” – once introduced as abbreviations, please ensure consistency in the use of these (e.g., RR for risk ratio). Please check your manuscript carefully for this. 

p.11: Under “Malignant disease”, please split the first sentence into two separate sentences starting the second one with “Data were extracted from all studies but Goldacre et al […]” and explain why the study of Goldacre et al was not included in your analysis.

p.12: “[…], while cervical malignancy risk was assessed one paper (Mikkelsen et al., 2019).” – change to “[…], while cervical malignancy risk was assessed in one paper (Mikkelsen et al., 2019).” 

p.13: Under “Malignant disease”, please check grammar and syntax of the last sentence/paragraph starting with “Three studies combined to assess [..]”.

p.14: “Future risk of long term depression, anxiety, stress and alcohol dependence were assessed after pregnancy loss in five studies of women with a history of miscarriage […] and in six studies in women with a history of stillbirth [..] respectively” – please change “long term” to “long-term” and remove “respectively”.

DISCUSSION

Please avoid assertions of primacy ("We report for the first time....").

Please present and organize the Discussion as follows: a short, clear summary of the article's findings; what the study adds to existing research and where and why the results may differ from previous research; strengths and limitations of the study; implications and next steps for research, clinical practice, and/or public policy; one-paragraph conclusion.

p.15: “Our results suggest that women experiencing stillbirth have an increased risk of developing ischaemic heart disease and cerebrovascular disease by 74 and 71% respectively” – please ensure to add units when necessary and check carefully throughout your manuscript.

p.16: “However, other risk factors such as such as hyperlipidaemia, hyperglycaemia, […]” – please remove the duplicate “such as”.

FIGURES

Please provide titles and legends for all figures (including those in Supporting Information files).

For all Figures, please ensure that you have complied with our figures requirements http://journals.plos.org/plosmedicine/s/figures.

Please consider avoiding the use of red and green in order to make your figure more accessible to those with colour blindness.

Please ensure to define abbreviations used in your figures and tables.

REFERENCES

PLOS uses the numbered citation (citation-sequence) method and first six authors, et al.

Comments from the reviewers:

Reviewer #1: The authors conducted a systematic review and meta-analysis of 56 studies on the association between pregnancy loss and long-term maternal morbidity and mortality. The authors found that women with a history of stillbirth increased risk of ischaemic heart disease, cerebrovascular disease, and renal disease compared with no stillbirth. Additionally, women with a history of stillbirth had a lower risk of breast cancer. The review is comprehensive, well-structured, and the conclusions are supported by the evidence. The article has already been revised once (I was not consulted at the initial revision).

There seem to be 2 issues in this paper:

-Definition of exposure - stillbirth bs miscarriage are not adequately defined in the methods or elsewhere. The definition provided in the introduction is not necessarily the one that will be used by many of the studies reviewed.

-Potential for confounding (stillbirth may be secondary to underlying morbidity). 

Introduction

-the authors give the definition of stillbirth from the WHO. However, most countries in this review will likely be from high income regions where the definition of stillbirth differs. In Canada, we use 20 weeks as the cutoff, same for the US.

Methods:

* Search Strategy (page 5): Provide a detailed search strategy that is transparent and repeatable, including the names of the databases searched, key words, search terms (both free-text and controlled vocabulary terms). Additionally, as the search was up to May 2022. This seems far back and could be updated to a more recent end date.

* Study selection (page 5): Consider providing the inclusion criteria.

* Data extraction (page 6): As the included studies were observational, it should be possible to extract the confounding factors for each study as well as the adjusted hazards ratios or risk ratios.

* Exposure (page 6): Miscarriage or stillbirth should be clearly defined, including gestational age cut-off.

* Outcomes (page 6): The outcomes could be better justified and organized. For example, "ischaemic heart disease (IHD), cerebrovascular disease, overall circulatory disease" - how about other cardiovascular diseases such as heart failure? What diseases were included in overall circulatory disease?

* Statistical analysis (page 6): Describe the statistical analysis method used for calculating a summary measure of effect size in a meta-analysis, such as the Mantel Haenszel, Peto, or other method. 

* As the included studies were observational, the authors could pool the adjusted hazards ratios or adjusted risk ratios adjusted for confounding factors, rather than crude risk ratios only. This would help as stillbirth may simply be a marker of underlying morbidity which is the cause of future diseases.

Results

-Is there a way to make the text more legible -page 8 is very hard to read.

-hard to interpret the findings without the definition of the exposure

Discussion

-Lots of potential for confounding due to unmeasured characteristics of women, especially comorbidity such as preeclampsia. This issue could receive better treatment in the discussion, and possibly even in the methods (produce adjusted pooled estimates).

Reviewer #2: See attachment

Michael Dewey

Reviewer #3: Thank you for the opportunity to review this interesting paper. The topic is important and the authors have gone to very considerable effort to consolidate and synthesise knowledge on this relatively overlooked area of research. I have a number of comments which may warrant consideration, particularly relating to the level of detail provided in the Methods section and how the Results are communicated.

Abstract:

Within the Methods of the Abstract, need to change "malignant" to "malignancy"

Within Results section of the Abstract, it may be easier - or more intuitive - to present the results in the sequence of cardio/cerebro-vascular disease, renal disease, malignancy (results of which are counter-intuitive) and mental illness. 

Introduction:

Third paragraph is a little difficult to read. I think this could be phrased more convincingly, the rationale needs to be somewhat stronger.

Methods:

In general, the Methods section is lacking in detail. Some of this may be due to word count consideration or the prior publication of a protocol on PROSPERO, but in its present state the Methods section does not provide sufficient detail for meaningful interpretation of the Results.

The authors state that the grey literature was searched "by manually screening the reference list of papers included in the review and reviewing the reference list of related published systematic reviews". I am not convinced that this approach would provide a comprehensive overview of relevant grey literature and I would suggest rephrasing here. To my mind, this might add a very limited number of studies or technical reports which have not been peer reviewed, and which happen to have been cited by others previously. But it is unlikely to provide a broad, comprehensive scoping overview of the grey literature.

There is insufficient detail given on what outcomes the authors sought, and how these were ascertained. For example, in terms of neoplastic outcomes, were benign tumours included? If so, were diagnoses of benign tumours potentially included within the same meta-analyses as mortality from metastatic cancer? (And if so, this would likely introduce an unacceptable degree of clinical diversity in to meta-analyses) How were decisions taken to differentiate outcome measures? Within the Results section it becomes more apparent that the authors focused particularly on breast, ovarian, endometrial and cervical cancer, but this is not clear from the Methods.

Further detail is required on methods of outcome ascertainment. Did the authors accept any self-reported diagnosis of an outcome, or was a doctor's diagnosis or laboratory/histological confirmation required? Or did they rely on ICD/DSM coding of relevant diagnoses? This detail is required for comparison with any future studies. If multiple methods of outcome ascertainment were accepted, ideally subgroup analyses might be conducted with this differentiation in mind.

There is insufficient detail provided on the eligibility of included studies on the basis of their inherent design (i.e. case control, cross-sectional, cohort). There is also insufficient detail provided on how pre-existing morbidity was considered in term of eligibility and/or subsequent analysis. For example, what did the authors accept as adjusted effect estimates from individual studies - was this based on a pre-determined minimal suite of essential confounders, or just based on the study's own definition of adjustment?

In Figure 1 (flow diagram) it states that 18 papers were not attainable, and were thus excluded. I would expect some detail in the Methods section regarding the steps taken to try to ensure that all eligible papers were retrieved. Perhaps this was done - but the detail is not provided here.

Results:

In Figure 1 it is stated that there were 1080 "additional records identified through other sources". I am not clear on what is meant by this. Are these all from reference lists of other articles? This seems like a very large number of articles to add to the flow diagram without a full explanation for their source.

The second paragraph of Results section is problematic for a few reasons: 

"The mean maternal age at the time of fetal loss was on average 23 years at the time of loss (Pell et al., 2004) and 69 years (Parker et al., 2014) at the time of outcome measure." Firstly, the study by Pell et al. (2004) is not within the list of References. Secondly, it is unclear from this sentence how these references should be applied here? Is the mean maternal age at the time of fetal loss not based on the entire pooled cohort of affected women across all eligible studies, and likewise for the age of outcome measure? If so, why would only one reference apply for each of these points? Moreover, the next sentence states "The mean follow-up period for outcome assessment for the miscarriage cohort was 16.9 while for stillbirth this was 16.4 years." If this is true, then one would imagine that the mean age at outcome measurement would be approximately 16-17 years after their most recent fetal loss (but perhaps not exactly so, depending on which studies reported this data). Based on this rationale, one might expect that women were, on average, aged 52-53 years at their most recent fetal loss (i.e. 69-16 = age 53 years) according to the description provided. Presumably this is incorrect. Maybe I have misinterpreted what the authors are trying to communicate here - but if so, I think there is scope for others to misinterpret this too. I think the phrasing needs to be improved throughout this paragraph.

Outcomes:

It is difficult to know how to interpret some of these findings without further detail on how the authors dealt with the issue of pre-existing maternal morbidity. Were all analyses based on women who had no prior diagnosis of the outcome in question (e.g. for IHD, were all women with pre-existing cardiovascular disease excluded at baseline? What about major risk factors for IHD e.g. pre-existing chronic hypertension?)

I think some subgroup analysis is required for some of the key results. For example, in figure 2ai, the largest studies all appear to show a statistically significant association between miscarriage and IHD, but there are multiple smaller studies which do not. The forest plot suggests a modest, but non-significant association overall, and the authors present this as a null association. I am unsure about this interpretation. Could it be an issue related to methodological diversity between included studies in the meta-analysis (based on sample size, or study design)? A similar issue arises for figure 2bi where the largest studies of miscarriage and stroke all appear to show a statistically significant association (perhaps based on greater statistical power?) but multiple smaller studies show a null association, and this is also what the forest plot concludes. I think subgroup analysis could help to explain whether true associations might exist or not.

Minor point, on page 12 the sentence "Moreover, endometrial (Mikkelsen et al., 2019) (Xu et al., 2004) and ovarian (Mikkelsen et al., 2019) malignancy risk was examined in two and one studies respectively, while cervical malignancy risk was assessed one paper (Mikkelsen et al., 2019)." appears twice.

The findings for Mikkelsen 2019 are counter-intuitive and seem to heavily influence the results of the meta-analysis of stillbirth and breast cancer (which feature prominently, including in the Abstract) and which is ultimately based on a small number of included studies (n=3). From a quick look at the original analysis by Mikkelsen et al. 2019, their main analysis reported a null association between stillbirth and breast cancer (rather than a seemingly protective effect). Perhaps the authors chose to compare number of outcomes among parous women only - and if so that's fine. But it would be helpful to state whether this approach was consistently applied across all studies (if so, this needs to be clearer in the Methods section). When the authors use the term "controls", do they mean parous women who never experienced pregnancy loss? Or do they mean all women (including those who were never pregnant) who have never experienced pregnancy loss? This distinction is important for the interpretation of the Mikkelsen 2019 data. I would also suggest making it clear that this result is based on a small number of studies and is very much dominated by the authors' interpretation of the data presented by Mikkelsen et al., otherwise there may be scope for misinterpretation of the clinical relevance of this.

Regarding mental health outcomes, these results are also difficult to interpret without some detail on outcome ascertainment and definitions of various outcomes (e.g. depression). Clearly, the meta-analysis findings are approaching statistical significance and thus should be interpreted with some caution. Again, is there scope for some subgroup analysis to aid interpretation here?

Discussion:

Several of the aforementioned points need to be addressed either within the body of the manuscript or within a much more lengthy section on Limitations.

The comparison of findings between this manuscript and the previous systematic review by Oliver-Williams et al. requires further elaboration. Are there other methodological differences which existed between these two systematic reviews? 

It would be helpful to have some further discussion on what the authors believe to be the implications of their findings, including consideration of the absolute risks to affected women if associations herein are true. Miscarriage is considerably more prevalent than stillbirth, and thus associations between stillbirth and maternal morbidity may be important in relative terms, but less concerning in absolute terms compared to even modest associations between miscarriage and maternal outcome measures.

[LINK]

---

## [Decision Letter · Decision Letter 2]

24 Nov 2023

Dear Dr. Daru,

Thank you very much for re-submitting your manuscript "Fetal loss and long-term maternal morbidity and mortality; a systematic review and meta-analysis." (PMEDICINE-D-23-00438R2) for review by PLOS Medicine.

Thank you for addressing the comments from the editors and reviewers in detail. I have discussed the paper with my colleagues and the academic editor, and it has also been seen again by three of the original reviewers. The changes made to the paper were mostly satisfactory to the reviewers, and there is a list of remaining comments of an editorial nature. As such, we intend to accept the paper for publication, pending your attention to the detailed comments below in one further revision. When submitting your revised paper, please include once again a detailed point-by-point response to the editorial comments.

[LINK]

We expect to receive your revised manuscript within 1 week. Please email me (aschaefer@plos.org) if you have any questions or concerns.

If you have any questions in the meantime, please contact me (aschaefer@plos.org) or the journal staff on plosmedicine@plos.org.  

We look forward to receiving the revised manuscript by Dec 01 2023 11:59PM.   

Sincerely,

Alexandra Schaefer, PhD

Associate Editor 

PLOS Medicine

plosmedicine.org

Requests from Editors:

ACADEMIC EDITOR COMMENTS

The authors have tried to address the issues as well as they can - however, I agree with the reviewer that the conclusions remain overstated, given the limitations of the study (albeit that limitations are now discussed more fully).

EDITORIAL COMMENTS

1) In line with the comments made by the Academic Editor and Reviewer #1, we suggest toning down the conclusions throughout the paper.

2) Similarly, in line with Reviewer #1 comments, we feel that that the influence of comorbidity should be better discussed/disentangled, particularly with regard to the potential for gestational diabetes and preeclampsia, both of which predispose to both stillbirth and CVD. We agree that this merits additional discussion in the paper regarding these potentially explanatory pathways with respect to CVD outcomes.

3) We concur with the statistical reviewer's suggestion to include prediction intervals alongside confidence intervals in the manuscript. Kindly integrate prediction intervals into your analysis.

4) The studies included are all observational – please state this in the abstract (also see comment 1) under ABSTRACT).

5) In the title, please exchange the semicolon with a colon.

6) Regarding presentation style (e.g., Abstract ‘Findings section’ or Discussion), we ask that you not write, e.g., “...had a 56% greater risk (RR 1.56 …); “had a greater risk (RR 1.56 …” is our preferred style. Similarly, phrases like “we were unable to show” should be exchanged with “there was no evidence of increased risk” or similar.

7) When revising your manuscript, please check carefully for punctuation.

ABSTRACT

1) Thank you for explaining that details about the types of study designs included, and synthesis/appraisal methods are stated within the Methods sections. Please also include these details in the Abstract.

2) Please combine the Methods section and Findings section into one Methods and Findings section.

3) l.51: Please fix “p<001”. 

4) l.57: Please include square/box brackets for the CI values.

5) In the last sentence of the Abstract Methods and Findings section, please describe the main limitation(s) of the study's methodology.

AUTHOR SUMMARY

1) Thank you for providing an Author Summary. Unfortunately, the current Author Summary does not conform to our requirements. The summary should include 2-3 individual bullet points (single sentences for each) under each of the questions (Why Was This Study Done?; What Did the Researchers Do and Find?; What Do These Findings Mean?). In the final bullet point of ‘What Do These Findings Mean?’, please describe the main limitations of the study in non-technical language.

Please see our author guidelines for more information: https://journals.plos.org/plosmedicine/s/revising-your-manuscript#loc-author-summary.

It may be helpful to review currently published articles for examples which can be found on our website here https://journals.plos.org/plosmedicine/

INTRODUCTION

1) l.81: Please define ‘ICD’ at first use.

2) l.86: The first quotation marks are missing, please add.

3) l.89: “Stillbirth is reported in more than 2,500,000 women annually.” – please add a reference.

4) ll.180-182: This sentence currently reads as if North America and Europe are high-income countries. Please re-phrase (e.g., ‘high-income regions/countries...’).

5) l.189: “(37/59)” – please change to, e.g. (n=37 out of 59) or (n=37).

6) Table 1: In the column ‘Duration of follow-up’, some numbers do not have units (years?, months?) – please revise.

7) Table 1: In the ‘Result’ column in brackets, one definition next to RR, OR etc. seems to be missing. For example, for the study of Andalib et al. (2006), Iran, the results seem to be presented as “n ± SD”.

8) Table 1: Please ensure that for all numbers in the ‘Result’ column it is evident what the numbers are (RR, OR etc.). For example, for Bergant et al. (1997), Austria, it is not clear. Also, when presenting 95% CI values, please ensure to add “95% CI” before the brackets.

9) Table 1 and Table 2: Please note the use of commas to separate upper and lower bounds, as opposed to hyphens as these can be confused with reporting of negative values.

10) Table 1: When stating age, please add a unit (e.g., for Toffol, Koponen, Partonen (a) (2013), Finland).

11) Table 1: Please define TIA.

12) Table 2: Please define HR.

13) Table 2: When presenting 95% CI values, please ensure to add “95% CI” before the brackets.

14) l.241: Is IHD correct here as the section discusses cerebrovascular disease including haemorrhagic and ischaemic stroke?

15) l.271: “[11,31,35]” seems to be misplaced – please revise.

16) l.288: “…without a history, of which 6,381 (2.78%) exposed and 23,514 (3.12%)” – The sentence seems to be incomplete, please revise.

17) l.299: “[20,44,57,60]” seems to be misplaced – please revise the placement of references through the entire main manuscript.

DISCUSSION

1) ll.393-395: Please provide references.

2) Please remove the ‘Conclusion’ subheading. The ‘Conclusion’ paragraph should be the last part of the Discussion section.

FIGURES

1) The Figures are of very low quality, please check.

2) Figure 1: In the ‘Records identified from:’ box, please name the specific databases with the according numbers of records identified.

3) Figure 2: In the Figure description, please add “versus” to “(bii) Cerebrovascular in women with stillbirths no stillbirth”. 

4) Figure 2 and Figure 3: Please revise the x-axis labels as it appears that on some graphs ‘Exposure [Increase Risk]’ and ‘Exposure [Decrease Risk]’ are on the wrong side.

5) Figure 2/Figure 3/Figure 4/Figure 5: Please delete the “year” column and only add the year to first column ‘Study or Subgroup’.

SUPPLEMENTARY MATERIALS

1) Please provide the completed PRISMA checklist and the Abstract PRISMA checklist as separate Supporting Information files and do not list them as S Figures.

2) Please ensure that you provide x-axis labels for all graphs (e.g. see S8 Fig (c)).

3) S8 Fig – S17 Fig: Please delete the “year” column and only add the year to first column ‘Study or Subgroup’.

4) Please ensure to define all abbreviations used in your supplementary figures and tables.

REFERENCES

1)Where website addresses are cited, please use ‘accessed’ instead of ‘cited’ when specifying the date of access.

2) Please thoroughly revise all references and ensure that journal name abbreviations match those found in the National Center for Biotechnology Information (NCBI) databases (http://www.ncbi.nlm.nih.gov/nlmcatalog/journals), and are appropriately formatted and capitalised (e.g., for reference [17] American Journal of Obstetrics and Gynecology should be Am J Obstet Gynecol).

SOCIAL MEDIA

To help us extend the reach of your research, please provide any X (formerly known as Twitter) handle(s) that would be appropriate to tag, including your own, your coauthors’, your institution, funder, or lab. Please respond to this email with any handles you wish to be included when we tweet this paper.

Comments from Reviewers:

Reviewer #1: I think that the revised version has improved. I selected "Proceed without recommendation" because I felt the overall conclusion was somewhat overstated. The problem is that it is very well known that gestational diabetes and preeclampsia are strong determinants of CVD. These pregnancy disorders are also strongly associated with stillbirth. Reading the discussion, it felt like not enough emphasis was put on these potential explanatory pathways (guidelines for CVD risk reduction include these two pregnancy conditions, so an argument for including stillbirth seems uncertain). However, the manuscript is also fine in this form if the editors are satisfied with the revision.

Thank you for consulting me.

Reviewer #2: The authors' rebuttal is fine but contains one statement which I find hard to accept.

I recommended also presenting prediction intervals as well as confidence intervals. The authors replied that

"... we want our results to be used in clinical settings showing the degree of uncertainty in our point estimates

rather in research setting where prediction of a subsequent study estimates may be important."

If a group of clinicians reads this review and asks themselves "What would happen here?" then that is exactly the same as asking what would happen in the next study and so for that purpose they need a prediction interval. Knowing the variance of the distribution of true effect sizes is a theoretical interest. So my feeling is that the authors have this the wrong way round.

Michael Dewey

Reviewer #3: I am satisfied that the authors have addressed my queries. Thank you for considering my previous feedback and for the opportunity to review this relevant and timely article.

[LINK]

---

## [Decision Letter · Decision Letter 3]

12 Dec 2023

Dear Dr. Daru,

Thank you very much for re-submitting your manuscript "Fetal loss and long-term maternal morbidity and mortality; a systematic review and meta-analysis." (PMEDICINE-D-23-00438R3) for review by PLOS Medicine.

Thank you for addressing the comments of the editors and reviewers in detail. I have discussed the paper with the academic editor, and it has also been reviewed again by the statistical reviewer. The changes made to the paper were satisfactory to the academic editor and the reviewer. However, there are a few inconsistencies in the figure numbering/description that will need to be addressed in one final revision, along with a few other editorial points that are outlined below. When submitting your revised paper, please include a detailed point-by-point response to the editorial comments.

[LINK]

We expect to receive your revised manuscript within 1 week. Please email me (aschaefer@plos.org) if you have any questions or concerns.

We look forward to receiving the revised manuscript by Dec 19 2023 11:59PM.   

Sincerely,

Alexandra Schaefer, PhD

Associate Editor 

PLOS Medicine

plosmedicine.org

Requests from Editors:

1) l.60: Please introduce the abbreviation “DM” before use (in line 48).

2) Author summary: Please re-phrase the first bullet point under ‘Why was this study done?’ and refrain from using evaluative language like ‘devastating’. Editorial suggestion: “…one of the most serious complications…”

3) Author summary: Please remove ‘across all body systems’ from the third bullet point ‘Why was this study done?’.

4) Author summary: Under ‘What do these findings mean?’, please change ‘type 2 diabetes’ to ‘type 2 diabetes mellitus’ (as described throughout the main text) and switch bullet point 2 and 3 so that the main limitation is the final bullet point.

5) l.96: Please note that references should be mentioned in one bracket, e.g. instead of “…before 20 weeks gestation [2] [3]..”, please write “…before 20 weeks gestation [2,3]..”. Please revise throughout the entire main manuscript.

6) l.178: Please introduce the abbreviation ‘PI’ following “…and 95% prediction intervals.”.

7) l.266 and following: You abbreviate type 2 diabetes mellitus from here on as T2DM whereas earlier you used the abbreviation “type 2 DM”. Please use a consistent format; we suggest using T2DM.

8) l.356: Please change to “…in this systematic review.”.

9) ll.393-395: Please provide references for both sentences.

10) The figure numbering, descriptions and the presentation within the manuscript does not match. Please thoroughly revise all of your figures for showing the correct graphs with correct labeling and ensure that the according figure descriptions match. I have included some specific issues in the points below (11-16).

11) Please use standard alphabetical labeling for all figure panels (i.e., a, b, c, etc.).

12) Figure 3: In the figure description, “type 2 diabetes mellitus” should be written in full first before introducing the abbreviation. Please revise.

13) Figure 3/4: In l.323 (“Analysis showed a RR of 1.17, (97% CI [0.95,1.45]; p=0.15 I2 = 0%, 95% PI [1.13, 1.29]) (Fig 4d).”), for example, the sentence refers to Figure 4d which should be Figure 3d. The figure description for Figure 3 lacks a description for 3c and 3d. Also, for Figure 3d, it seems the x-axis labels (“[Increased risk]” and “[Decreased risk]”) need to be the other way around (i.e., women with a previous history of stillbirth had a 1.17 increased risk of all cancer-related mortality).

14) In the Figure description of Figure 4, you use the label “(c)” twice and on the graph itself both graphs do not exist (are these graphs 3c and 3d?). Please thoroughly revise.

15) In the Figure description of Figure 5 you label the different graphs as (ai) and (bi) whereas on the graph itself these are labelled (ai) and (aii). Please revise and see comment #11.

16) Please remove the “year” column from Figure 4 (ai).

17) Please re-name the individual titles of the supplementary figures and table including a “S” (as done in the overview of the Supporting Information). For example, on page 3 “Fig 1: Keyword Strategy” should be “S1 Fig: Keyword Strategy”. Also, S1-S4 Fig should be named tables instead of figures.

18) The figure title/description of S7 Fig in the overview does not match the description above the graphs. It seems S7 and S8 need to be swapped in the overview list.

19) It seems that you skipped “Fig 12” when numbering your supplementary figures and wrongly labelled it “Fig 13” which should be “S12 Fig”. Followingly, “Fig 14” should be “S13 Fig” and so on. Please revise.

Comments from Reviewers:

Reviewer #2: The authors have addressed my remaining point.

Michael Dewey

[LINK]

---

## [Editor Report · Decision Letter 4]

3 Jan 2024

Dear Dr Daru, 

On behalf of my colleagues and the Academic Editor, Sarah Stock, I am pleased to inform you that we have agreed to publish your manuscript "Fetal loss and long-term maternal morbidity and mortality; a systematic review and meta-analysis." (PMEDICINE-D-23-00438R4) in PLOS Medicine.

Thank you for your considered and detailed responses to the comments provided by the editors and reviewers throughout the editorial process. We look forward to publishing your manuscript, and editorially there are only a few remaining minor stylistic/presentation points that should be addressed prior to publication. We will carefully check whether the changes have been made. If you have any questions or concerns regarding these final requests, please feel free to contact me at aschaefer@plos.org.

We noted a few remaining inconsistencies in the figure numbering/description. Please once again thoroughly revise the figure numbering and their mentioning throughout the main body of the manuscript. Some specific issues are outlined below.

1) l.304: “The pooled RR was 1.00 (95% CI [0.94,1.06]; p=0.98 I2 = 0%, 95% PI [0.78,1.57]) (Fig 3c).” – Figure 4c should be mentioned here instead of 3c. 

2) ll.314-315: “The pooled risk of the included studies demonstrated a small but significant reduction in the risk of breast cancer (RR: 0.80, 95% CI [0.67,0.96]; p=0.02 I2 = 0%, 95% PI [0.72,0.93]) (Fig 3b).” - Figure 4b should be mentioned here instead of 3b.

3) l.322: “Analysis showed a RR of 1.17, (97% CI [0.95,1.45]; p=0.15 I2 = 0%, 95% PI [1.13, 1.29]) (Fig 3e).” – Figure 4e should be mentioned here instead of 3e.

PRESS

Sincerely, 

Alexandra Schaefer, PhD 

Associate Editor 

PLOS Medicine